# Star Elastic: Many-in-One Reasoning LLMs with Efficient Budget Control

Ali Taghibakhshi [* 1]  Ruisi Cai [* 1 2]  Saurav Muralidharan [* 1]  Sharath Turuvekere Sreenivas [* 1]
Ameya Sunil Mahabaleshwarkar [1]  Marcin Chochowski [1]  Akhiad Bercovich [1]  Ran Zilberstein [1]  Ran El-Yaniv [1]
Yonatan Geifman [1]  Daniel Korzekwa [1]  Yoshi Suhara [1]  Oluwatobi Olabiyi [1]  Ashwath Aithal [1]  Nima Tajbakhsh [1]
Pavlo Molchanov [1]

## Abstract

Training a family of large language models (LLMs), either from scratch or via iterative compression, is prohibitively expensive and inefficient, requiring separate training runs for each model in the family. In this paper, we introduce **Star Elastic**, a novel LLM post-training method that adds $N$ nested submodels to a given parent reasoning model using the compute of one run ($N\times$ savings) via a single post-training job. Beyond reducing training costs, Star Elastic also addresses a fundamental limitation in efficient reasoning: the rigidity of static architectures, which forces the allocation of constant resources regardless of token difficulty. By unlocking elastic budget control, Star Elastic enables a novel approach that uses different submodels for each reasoning phase (thinking and answering). Star Elastic supports (1) nesting along the SSM, embedding channel, MoE and FFN axes, (2) learning nested submodels via an end-to-end trainable router, and (3) curriculum-based knowledge distillation. We apply Star Elastic to the NVIDIA Nemotron Nano models; in particular, we demonstrate its effectiveness on hybrid MoE architectures with Nemotron Nano v3 (30B/3.6A), generating 23B (2.8A) and 12B (2.0A) variants with 160B training tokens. For Nemotron Nano v2 (12B), we produce 9B and 6B nested models using only 110B training tokens, achieving a $360\times$ reduction versus training from scratch and a $7\times$ reduction over state-of-the-art compression methods. All nested models match or outperform independently trained baselines of comparable size. Crucially, elastic budget control advances the accuracy–latency Pareto

frontier, achieving up to 16% higher accuracy and $1.9\times$ lower latency via dynamic per-phase model selection.

## 1. Introduction

Large Language Model (LLM) families typically consist of a set of fixed-size models, offering users discrete accuracy–cost trade-offs across diverse deployment settings (Touvron et al., 2023; Dubey & et al, 2024). However, each model in the family is typically trained independently and stored as a separate set of parameters, making this approach prohibitively expensive in terms of compute/storage resources, and deployment overhead. For example, the Llama-3.1 family spans 8B, 70B, and 405B parameters (Dubey et al., 2024), with each variant trained from scratch on tens of trillions of tokens, multiplying training and storage costs, and restricting users to a small set of predefined model sizes. Recent model compression methods mitigate the cost of training large language model (LLM) families through structured pruning and knowledge distillation, training only the largest model from scratch and deriving smaller variants through compression (Muralidharan et al., 2024; Xia et al., 2023). While effective, these approaches still require hundreds of billions of training tokens per compressed model, keeping overall training costs high. *Elastic* nested models, on the other hand, embed multiple nested sub-models within a single parent model, enabling zero-shot extraction of multiple model sizes from one training run and facilitating efficient multi-size deployment (Cai et al., 2024; Kudugunta et al., 2023).

In parallel, hybrid LLMs combining attention, State Space Models (SSMs, such as Mamba), MLPs, and Mixture-of-Experts have become popular (Gu & Dao, 2023; Dao & Gu, 2024; Lieber et al., 2024; Glorioso et al., 2024; Blakeman et al., 2025a), offering improved efficiency through reduced KV cache, linear-time sequence processing, and conditional expert computation, while maintaining strong accuracy. Unfortunately, no current framework or model supports elasticity (nesting) for such hybrid LLMs; the limited efforts targeting compression of such models (Shukla

---

[*]Equal contribution [1]NVIDIA Corporation [2]Work done during an internship at NVIDIA. Correspondence to: Ali Taghibakhshi <ataghibakhshi@nvidia.com>.

*Proceedings of the 43rd International Conference on Machine Learning*, Seoul, South Korea. PMLR 306, 2026. Copyright 2026 by the author(s).

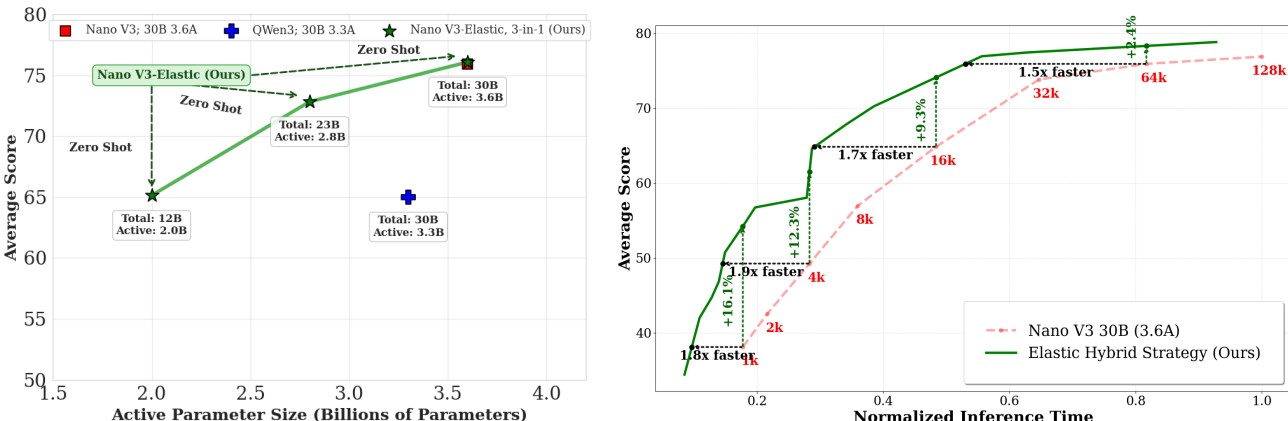

*Figure 1.* **Left**: Average accuracy of Nano v3 Elastic compared to parent and Qwen models across key reasoning benchmarks (AIME-2025, IFBench, GPQA, LiveCodeBench v5, and MMLU-Pro). **Right**: Hybrid elastic budget control versus standard Nemotron Nano v3 reasoning budget control; Star Elastic improves the accuracy–speed Pareto frontier. Time is measured using vLLM.

et al., 2024) that we know of do not support heterogeneous expert or FFN channel selection.

In this paper, we introduce **Star Elastic**, a novel post-training approach for hybrid Mamba–Transformer–MoE LLMs that produces multiple nested sub-networks at different parameter budgets from a single training run, and supports elastic budget control during inference. Our approach combines (1) importance-based ranking of embedding channels, attention and SSM heads, MoE experts, and FFN channels (2) a learnable router that automatically determines nested submodels and employs knowledge distillation for joint sub-network optimization, (3) a two-stage training curriculum for optimal reasoning performance, and (4) elastic budget control allowing dynamic per-phase model allocation.

We apply **Star Elastic** to Nemotron Nano v3 MoE (30B/3.6A), producing 23B (2.8A) and 12B (2.0A) variants, and Nemotron Nano v2 (12B), producing 9B and 6B nested models with approximately 160B and 110B training tokens, respectively. All nested models match or outperform independently trained baselines, while enabling a $360\times$ token reduction compared to training from scratch and a $7\times$ reduction over state-of-the-art compression (Figure 5). Crucially, elastic budget control enabled by Star Elastic advances the Pareto frontier, achieving up to 16% higher accuracy and $1.9\times$ lower latency via dynamic per-phase model selection as shown in Figure 1.

**Key contributions:**

- Introduces the first elastic post-training method for **reasoning LLMs** with hybrid **Mamba–Attention–MoE** architectures.

- Demonstrates **elastic budget control**, dynamically al-

locating submodels across reasoning and answer generation phases, achieving up to 16% higher accuracy and $1.9\times$ lower latency.

- Introduces a **learnable router** that automatically determines nested submodel architectures, optimized through **knowledge distillation** from the parent model.

- Achieves **significant training cost reductions** for model families: up to $7\times$ over prior compression methods and $360\times$ over pretraining from scratch.

## 2. Elastic Model Construction

In this section, we describe how Star Elastic converts an existing LLM into an elastic model. The pipeline consists of three stages: (1) importance estimation to rank model components (embedding dimensions, attention and SSM heads, MoE experts, and FFN channels), (2) using an elastic formulation to enable nesting along both width and depth axes, and (3) elastic training using a two-stage curriculum with a learnable router. Figure 2 showcases the Star Elastic pipeline.

### 2.1. Importance Estimation and Model Preparation

The first stage of elastification involves computing the importance score for each model component, such as attention heads, FFN channels, layers, etc. Components are then sorted in decreasing order of importance, yielding ranking permutations $\sigma^{(w)}$ that reflect their contribution to model accuracy. These rankings guide the router's architecture selection during elastic training.

**Width:** In this work, we extend activation-based importance scoring (Muralidharan et al., 2024; Taghibakhshi et al., 2025) to hybrid Mamba-Attention-MoE architectures, and

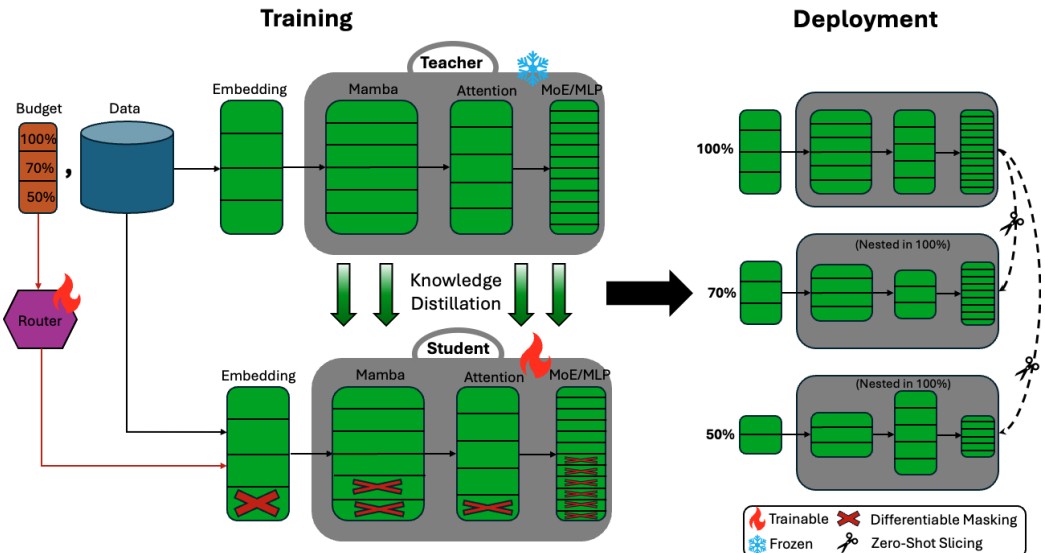

*Figure 2.* **Overview of the Star-Elastic training and deployment pipeline. Training:** For each training sample, data flows to both teacher and student models. A budget (e.g., parameter count, memory, or latency target) is selected and passed to the router, which generates differentiable masks for the student model. Knowledge distillation from the model prior to elastification enables simultaneous optimization across all budget variants. **Deployment:** After training, all models are extracted zero-shot from a single elastic checkpoint: the full model and nested sub-networks are immediately available without additional training. (Appendix F.2 contains deployment details).

compute importance scores for MoE experts, embedding channels, Mamba heads and head channels, attention heads and FFNs.

**Embedding channels:** We aggregate normalized input activations across sequence ($L$) and batch ($B$) dimensions to compute the importance score for each embedding channel $i$ as $F_i^{emb} = \sum_{B,L} |\text{LN}(X)|_i$.

**FFN and MoE channels:** Importance score for channel $i$ is computed from the activations of the first up-projection layer, summed over batch ($B$) and sequence length ($L$) as $F_i^{ffn} = \sum_{B,L} |X(W_{up})^T|_i$.

**Mamba heads and channels:** Following group-aware constraints that preserve SSM structure (Taghibakhshi et al., 2025), we compute head channel scores based on mamba input projection layer activations as $F_d^{head\_channel} = \left\| \sum_{B,L} s_{:,d} \right\|_2$, where $s = \text{LN}(X)(W_d)^T$. These scores are then aggregated across all heads. Individual head scores are computed using top-ranked channel activations ($\mathcal{D}_{top}$), as $F_h^{head} = \|s_{h,\mathcal{D}_{top}}\|_2$, where $h \in \{1, \ldots, m_h\}$ and $m_h$ denotes the number of mamba heads.

**Attention heads:** Importance for each head is computed from the attention scores aggregated across batch ($B$) and sequence length ($L$) as $F_h^{attn} = \sum_{B,L} \|Attn(XW_h^Q, XW_h^K, XW_h^V)\|_2$.

**MoE expert importance (REAP):** For MoE layers with conditional expert activation, we adopt Router-Weighted Ex-

pert Activation Pruning (REAP) (Lasby et al., 2025), which measures each expert's direct contribution to the layer output magnitude. The saliency score for expert $f_j$ is computed as $S_j = \frac{1}{|\mathcal{X}_j|} \sum_{x \in \mathcal{X}_j} g_j(x) \cdot \|f_j(x)\|_2$, where $\mathcal{X}_j$ is the set of inputs where $g_j(x) \in \text{TopK}(g(x))$, and $g_j(x)$ is the router gate-value for expert $j$. This criterion approximates expert importance by averaging the product of routing weights and expert output magnitudes over tokens where the expert is active. Unlike naive frequency-based pruning, REAP considers both routing frequency and functional contribution, enabling principled expert selection for heterogeneous MoE elastification.

**Depth:** Layer importance is estimated iteratively using normalized mean squared error (MSE) between the full model's logits and those obtained with individual layers removed. At each iteration, for every remaining layer $j$, we compute $s_j = \frac{\sum_{B,L}(\mathcal{M}_{full} - \mathcal{M}_{-j})^2}{\sum_{B,L} \mathcal{M}_{full}^2}$, where $\mathcal{M}_{full}$ denotes logits from the full model and $\mathcal{M}_{-j}$ denotes logits with layer $j$ removed in addition to previously pruned layers. The iterative procedure accounts for changes in layer importance conditioned on previously pruned layers (Muralidharan et al., 2024; Taghibakhshi et al., 2025). Normalization ensures that scores are comparable across different calibration datasets.

### 2.2. Elastic Formulation

We describe our nested, weight-shared architecture, which allows a single hybrid Mamba–Attention–MoE LLM to

adapt dynamically to varying resource constraints. The model can be resized along both width and depth, enabling instant extraction of sub-networks with different parameter budgets and active expert counts without additional fine-tuning.

**Elastic Width:** We define elastic choices for each axis and sub-network $\mathcal{S}$: embedding dimension $d_e^{\mathcal{S}}$, attention heads $n_h^{\mathcal{S}}$, Mamba heads $m_h^{\mathcal{S}}$ and head channels $m_d^{\mathcal{S}}$, MoE expert count $e^{\mathcal{S}}(\ell)$, and FFN intermediate dimension $f^{\mathcal{S}}(\delta)$. Sub-networks are constructed by selecting values from these axes according to a target budget. The FFN and MoE components use layer-specific indices, with $1 \leq \delta \leq N_F$ and $1 \leq \ell \leq N_{\text{MoE}}$, since their widths can be chosen independently per layer to support heterogeneous configurations, where $N_F$ and $N_{\text{MoE}}$ denote the number of FFN and MoE layers, respectively. In all cases, elastic choices form nested hierarchies: for each axis, we select a subset of components ranked by importance, such that smaller-budget sub-networks $\mathcal{S}$ always use the most salient contiguous components retained by larger-budget variants.

**Elastic Depth:** For the depth axis, elasticity is controlled by a binary selection vector $\gamma^{\mathcal{S}} = [\gamma_0^{\mathcal{S}}, \gamma_1^{\mathcal{S}}, \ldots, \gamma_{N-1}^{\mathcal{S}}]$, where $\gamma_i^{\mathcal{S}} \in \{0, 1\}$ indicates whether layer $i$ is active in sub-network $\mathcal{S}$. The choice of which layer to disable is determined by its importance score, with less important layers removed first.

**Implementation.** Full implementation details, including elastification of each axis, masking, and zero-shot slicing, are provided in Appendix F.

## 2.3. Elastic Training

**Router architecture and design:** We train dedicated routers to learn the optimal model-specific mapping between the user-provided budget and the corresponding value for each nested axis. Each router consists of two fully connected layers with leaky ReLU activation applied between them.

The input to the router, for each axis, is a one-hot encoded vector representing the target compression level (budget specification): $u^{(\text{axis})} = \zeta_l \in \mathbb{R}^{n_{\text{targets}}}$, where $\text{axis} \in (d_e, m_h, m_d, n_h, e, f)$, $\zeta_l$ is the $l$-th standard basis vector, and $n_{\text{targets}}$ is the number of target model configurations. For example, with three budgets (100%, 70%, 50%) as illustrated in Figure 2, the inputs are $u = [1, 0, 0]$, $[0, 1, 0]$, and $[0, 0, 1]$ respectively.

Each router is parameterized as: $h^{(\text{axis})} = \text{LeakyReLU}(W_1^{(\text{axis})} u^{(\text{axis})} + b_1^{(\text{axis})})$ followed by $z^{(\text{axis})} = W_2^{(\text{axis})} h^{(\text{axis})} + b_2^{(\text{axis})}$ where $W_1^{(\text{axis})} \in \mathbb{R}^{d_{\text{router}} \times n_{\text{targets}}}$ and $W_2^{(\text{axis})} \in \mathbb{R}^{n_{\text{out}}^{(\text{axis})} \times d_{\text{router}}}$ are

learnable parameters.

For heterogeneous configurations, the router outputs target size per layer: $n_{\text{out}}^{(\text{moe\_expert, het})} = |\mathcal{E}| \times N_{\text{MoE}}$; $n_{\text{out}}^{(\text{ffn, het})} = |\mathcal{F}| \times N_F$, where $|\mathcal{E}|$ and $|\mathcal{F}|$ denote the cardinality of target expert counts and FFN dimension sets.

**Loss formulation:** The router outputs are passed through Gumbel-Softmax with temperature $\tau$ to produce soft probability distributions $P_i^{(\text{axis})} = \dfrac{\exp\left(\frac{\kappa \log \pi_i^{(\text{axis})} + g_i}{\tau}\right)}{\sum_j \exp\left(\frac{\kappa \log \pi_j^{(\text{axis})} + g_j}{\tau}\right)}$ over configuration choices for each elastic axis, where $g_i \sim \text{Gumbel}(0, 1)$; $\kappa$ is the scaling factor to balance the relative magnitude of logits; $\tau$ is a temperature parameter that controls the smoothness of the approximation, and as $\tau \to 0$, the $P^{(\text{axis})}$ distribution approaches a one-hot vector.

At each training iteration, we sample from these distributions to obtain relaxed discrete selections that enable gradient flow to the router parameters. The router is trained to optimize a resource-aware objective that maps selected configurations to hardware and computational constraints: $\mathcal{L}_{\text{router}} = \|\mathcal{C} - \hat{\mathcal{C}}\|$, where $\mathcal{C}$ is the resource cost of configuration (parameter count, memory usage, latency, or throughput) chosen by the router and $\hat{\mathcal{C}}$ is the target resource cost.

**Knowledge distillation and multi-budget optimization:** Following Minitron (Muralidharan et al., 2024), Star Elastic leverages knowledge distillation (Hinton et al., 2015) exclusively during elastic training, using the non-elastified parent model as the teacher to guide both architecture choice and accuracy optimization through teacher-aligned signals. We compute $\mathcal{L}_{\text{KD}} = D_{\text{KL}}(p_\varphi(x; \tau) \| p_\theta(x; \tau))$,

where $p_\varphi(x; \tau)$ denotes the teacher's softmax output at temperature $\tau$, and $p_\theta(x; \tau)$ denotes the student elastic model's corresponding distribution.

The final objective combines $\mathcal{L}_{\text{KD}}$ with a router based loss: $\mathcal{L}_{\text{total}} = \mathcal{L}_{\text{KD}}(\theta) + \lambda \cdot \mathcal{L}_{\text{router}}(\psi)$, where $\psi$ denotes router parameters, and $\lambda > 0$ balances model accuracy against resource constraints.

This end-to-end optimization enables the router to make architecture choices directly from the actual training signal, rather than relying on zero-shot proxy metrics evaluated post-hoc–a key distinction from prior methods that decouple architecture configuration from training objectives. Our approach prioritizes sub-networks with strong capacity for continued learning under knowledge distillation, rather than those that simply minimize loss at initialization.

**Two-stage training with curriculum-based sampling:** Multi-budget elastic training (i.e., jointly training multiple nested models) requires carefully orchestrated data alloca-

tion across budget targets to prevent training imbalance and maintain accuracy across all sub-networks. Empirically, naïve uniform sampling in extended-context regimes causes the full-budget model to degrade while smaller budgets improve, indicating gradient competition and motivating our curriculum-based sampling strategy (see Appendix C.2). Hence, we introduce a two-stage training pipeline that adapts the sampling strategy to context length:

*Stage 1: Uniform Budget Sampling (Short Context).* During the initial short-context phase (sequence length $L_1 \approx 8192$, total tokens $T_1$), we employ uniform budget sampling. For $n_b$ target budgets, each training batch receives equal allocation with probability $\frac{1}{n_b}$. Uniform sampling ensures the router receives balanced training signals from all sub-networks, allowing architecture exploration without budget-specific bias.

*Stage 2: Curriculum-Based Non-Uniform Sampling (Extended Context).* During extended-context training (sequence length $L_2 = 49152$, total tokens $T_2$), we transition to non-uniform sampling that prioritizes full-budget models: $p_2(b) = \alpha_b, \quad \forall b \in \{1, \ldots, n_b\}$, where $\sum_{i=1}^{n_b} \alpha_i = 1$ and weights are typically skewed toward larger budgets.

**The role of extended-context training for reasoning:** Reasoning tasks require extended token budgets for multi-step inference, making short-context elastic training insufficient for developing true reasoning capability. Training with 49K-token contexts exposes elastic variants to long reasoning chains, motivating our two-stage curriculum that first recovers accuracy for smaller models in the family and then incorporates reasoning-specific abilities (refer to Appendix C.1 for details).

## 3. Elastic Budget Control

Existing budget control methods, such as the ones used in the NVIDIA Nemotron Nano models (Nano, 2025; Blakeman et al., 2025b), work in two phases: (1) **thinking phase**: generation starts by prompting the model to produce reasoning text via a pre-pended `<think>` token. The number of generated thinking tokens is monitored and the phase is terminated once the predefined budget (e.g., $2k$ tokens) is reached; (2) **answering phase**: the model is then forced to generate the final answer conditioned on the (potentially incomplete) reasoning text.

Unfortunately, existing budget control frameworks are forced to rely on static model architectures; i.e., they use the exact same model across prefill, thinking and answering. This results in computational inefficiencies when uniform capacity is applied to phases of heterogeneous complexity. In practice, reasoning and answer synthesis often exhibit disparate computational demands. This discrepancy moti-

vates *elastic budget control*: a mechanism to dynamically calibrate model capacity for each inference phase, aligning resource allocation with task requirements to achieve the optimal performance-efficiency trade-off.

Specifically, using two model variants of differing sizes - denoted as $\mathcal{M}_L$ (Large) and $\mathcal{M}_S$ (Small) - we evaluate four experimental scenarios:

- $\mathcal{M}_L \rightarrow \mathcal{M}_L$: $\mathcal{M}_L$ used for both phases.
- $\mathcal{M}_S \rightarrow \mathcal{M}_S$: $\mathcal{M}_S$ used for both phases.
- $\mathcal{M}_L \rightarrow \mathcal{M}_S$: $\mathcal{M}_L$ for thinking, $\mathcal{M}_S$ for answering.
- $\mathcal{M}_S \rightarrow \mathcal{M}_L$: $\mathcal{M}_S$ for thinking, $\mathcal{M}_L$ for answering.

Guided by the empirical findings detailed in Section 4.3, we identify $\mathcal{M}_S \rightarrow \mathcal{M}_L$ as the optimal configuration for elastic budget control. This design choice is rooted in the asymmetric computational requirements of the two phases: (1) **high-volume reasoning**: the thinking phase benefits from a larger token budget to explore complex reasoning paths. By utilizing $\mathcal{M}_S$, we can generate extensive reasoning traces with minimal computational overhead; (2) **high-fidelity synthesis**: the answering phase requires superior instruction-following and internal consistency to transform the reasoning trace into a correct final response. $\mathcal{M}_L$ provides the necessary cognitive capacity to ensure this synthesis is robust.

**Cache state sharing:** In our elastic budget control experiments, we align with the standard inference practice of recomputing cache states when switching between nested models. This strategy is consistent with recent approaches adopted in the Nemotron Nano models (Nano, 2025; Blakeman et al., 2025b) and Qwen3 (Yang et al., 2025), reflecting the current constraints of inference frameworks like vLLM and TensorRT-LLM.

While naive budget control (using a single fixed model) naturally allows for cache reuse, switching between a smaller "thinking" model and a larger "answering" model typically breaks cache compatibility. Star Elastic mitigates this issue by preserving the structure of Mamba and attention layers during elastification, maintaining cache compatibility across nested models. Empirically, we find cache states to be highly consistent across nested models, indicating that cache states can be safely transplanted between nested models (see Tables 14 and 15 in Appendix E). As a result, although our reported wall-clock times currently include cache recomputation overhead, our architecture enables seamless cache reuse once framework support becomes available. Consequently, our reported performance should be viewed as a conservative lower bound, with further gains expected as cache reuse is enabled.

# 4. Experiments and Results

We apply our Star Elastic method to two model families: (1) Nemotron Nano v3 30B3.6A (Blakeman et al., 2025b) which is a hybrid Mamba-Transformer-MoE model, and Nemotron Nano v2 12B (Nano, 2025), a hybrid Mamba-Transformer model.

## 4.1. Experimental Setup

**Nested compression:** For both Nano v3 and v2, we simultaneously train three nested models from a single parent architecture using multi-budget elastic compression: this includes the original parent model, and two smaller models. For Nano v3, the router optimizes for target active parameter budgets of 3.6B, 2.8B, and 2.0B, deriving 23B (2.8A) and 12B (2.0A) nested variants from the 30B (3.6A) parent. For Nano v2, the router optimizes for total parameter budgets of 12B, 9B, and 6B, simultaneously deriving 9B and 6B nested variants. The frozen version of the parent model (prior to elastification) serves as the teacher for knowledge distillation. Detailed architecture specifications for the router-selected configurations are provided in Appendix A.

**Dataset:** For importance estimation and knowledge distillation, we use the open-source data used to train the Nano v3 and v2 parent models (Blakeman et al., 2025b; Nano, 2025). Ablation studies on data blends are provided in Appendix C.3.

**Evaluation benchmarks:** We evaluate across a comprehensive suite of reasoning and knowledge benchmarks: MMLU-Pro (Wang et al., 2024) (college-level multiple-choice reasoning), GPQA (Rein et al., 2023) (graduate-level science questions), AIME-2024 and AIME-2025 (Mathematical Association of America, 2024) (invitational mathematics), LiveCodeBench v5 (Jain et al., 2024) (code generation), IFBench (Pyatkin et al., 2025) (instruction following), and Tau Bench (Barres et al., 2025) (industry-specific reasoning across airline, retail, and telecom domains).

**Hyperparameters and training setup:** For importance estimation, we use 1024 calibration samples with sequences of length 8192. Knowledge distillation is performed in two phases: Nano v3 Elastic is trained for ~100B tokens (batch 6144, sequence length 8k), followed by ~60B tokens (batch 1024, sequence length 49k), totaling 160B tokens; Nano v2 Elastic is trained for ~65B tokens (batch 1536, sequence length 8k), followed by ~45B tokens (batch 512, sequence length 49k), totaling 110B tokens.

**Optimizer settings:** Model parameters are trained with learning rates of $10^{-4}$ (Nano v3) and $9 \times 10^{-5}$ (Nano v2), while the router uses $10^{-2}$ for both. A 60-step linear

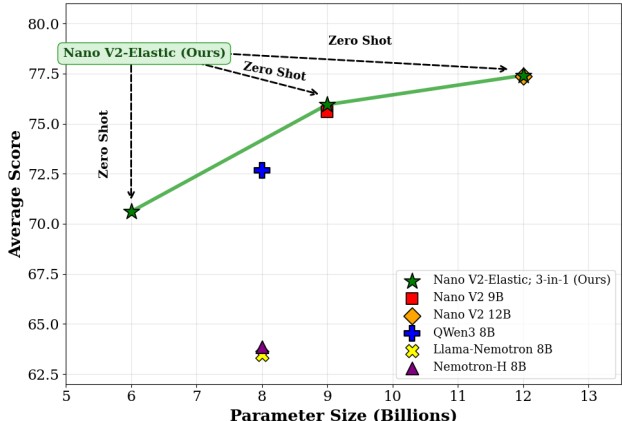

*Figure 3.* Average accuracy of Nano v2 Elastic across reasoning benchmarks detailed in Appendix B.

warmup is applied to all parameters. The Gumbel-Softmax temperature $\tau$ is annealed from 1.0 to 0.05. The router loss coefficient $\lambda$ is set to 1.0, and $\kappa$, the linear scaling coefficient for router logits, is scaled linearly from 1.0 to 10.0. The router intermediate hidden dimension ($d_{\text{router}}$) is set to 256.

**Budget sampling strategy:** During the short-context phase, we employ uniform budget sampling with $p(\text{budget}) = 1/3$ for each model variant. In the extended-context phase, we transition to weighted non-uniform sampling: $p(12\text{B}) = 0.5$, $p(9\text{B}) = 0.3$, $p(6\text{B}) = 0.2$ for Nano v2, with analogous weighting for Nano v3 variants. This prevents accuracy degradation with the full model during extended-context training. Ablations and details on the budget sampling strategy are discussed in Appendix C.2.

## 4.2. Main Results

Detailed accuracy results for the elastic Nano v3 and v2 variants is shown in Tables 1 and 7 (Appendix), respectively, with average scores in Figure 1 (Nano v3) and Figure 3 (Nano v2). Nano v3 Elastic-30B and Nano v2 Elastic-12B match the accuracy of their parent models, while smaller nested variants remain highly competitive against similarly-sized community models. Two-stage training with adjusted budget sampling prevents accuracy degradation in larger models (Table 9). Overall, Star Elastic scales effectively across model families and architectures, producing elastic models comparable to or surpassing independently trained counterparts with a fraction of the training budget (Figure 1 left and Figure 3).

## 4.3. Star-Elastic Budget Control Results

As described in Section 3, Star Elastic enables the use of different nested models during distinct reasoning stages such

*Table 1.* Detailed Star Elastic results on Nano v3. All three Elastic variants are obtained from a single 160B-token training run. **\*** Indicates the teacher model used for distillation.

| Benchmark | NanoV3 Elastic-12B (2.0A) | NanoV3 Elastic-23B (2.8A) | NanoV3 Elastic-30B (3.6A) | NanoV3-30B* (3.6A) | Qwen3-30B-A3B (3.3A) |
|---|---|---|---|---|---|
| AIME-2025 | 78.54 | 85.63 | **88.54** | 87.92 | 80.00 |
| GPQA | 57.39 | 69.82 | 72.10 | **73.11** | 70.83 |
| LiveCodeBench v5 | 55.24 | 67.30 | **72.70** | 71.75 | 68.25 |
| MMLU-Pro | 68.28 | 76.07 | 78.63 | 78.86 | **81.11** |
| IFBench (prompt) | 64.03 | 67.43 | **70.58** | 70.82 | 43.28 |
| IFBench (instruct) | 67.39 | 70.75 | **73.96** | 73.19 | 46.57 |
| Tau-Airline | 24.67 | 38.67 | 43.33 | 44.67 | **52.67** |
| Tau-Retail | 49.12 | 55.56 | **59.36** | 53.51 | 56.43 |
| Tau-Telecom | 29.33 | 30.99 | **33.33** | 30.99 | 28.36 |

as thinking and answer generation. Figure 1 (right) shows the results for our proposed elastic budget control method and compares it to the default Nano v3 budget control. We notice from the Figure that our approach achieves accuracy improvements up to 16% and latency reductions of up to 1.9×. Figure 4 summarizes the elastic budget control configurations underlying the accuracy–speed Pareto improvements in Figure 1 (right). We compare hybrid thinking–answering allocations across models and observe that the $\mathcal{M}_S \to \mathcal{M}_L$ strategy (small thinking, large answering), specifically in the case of 23B → 30B, provides the best accuracy–latency tradeoffs over a wide range of budgets, since it concentrates capacity on the final answer while keeping the reasoning phase cheaper. For applications that require the highest absolute accuracy and can afford more inference time, the 30B → 23B configuration is preferred, slightly outperforming other settings at the high-latency end of the frontier. At the lowest budgets, either 12B → 30B or 30B → 12B can be used to minimize latency while still benefiting from asymmetric thinking–answering allocations. To measure inference latency, we utilized the NeMo-Skills library[1] with vLLM as the backend, employing BF16 precision. We set the batch sizes to 40, 30, and 16 for the 12B, 23B, and 30B models, respectively. Detailed per-benchmark Pareto curves and configuration-level analyses, including a corresponding scatter plot for all combinations of thinking and answering, are provided in Appendix D.

## 4.4. Runtime Speedups

Table 2 shows the throughput improvements that Nano v3 Elastic models achieve at an input and output sequence length of 8192 and 16384, respectively; we use the maximum batch size that fits each model on an H100 GPU at bfloat16 precision. We notice that Star Elastic nested models achieve significant speedups (up to 2.4× in this case) while also enabling inference at much higher batch sizes on the same GPU.

[1]https://github.com/NVIDIA-NeMo/Skills

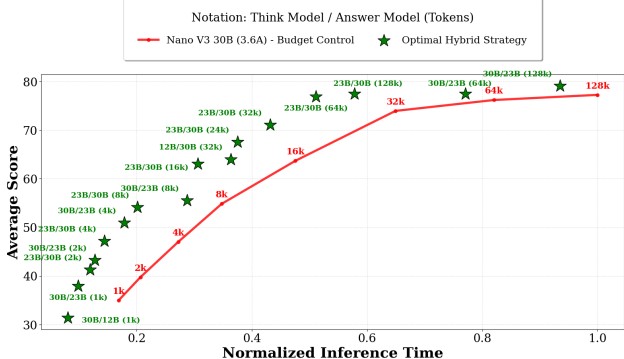

*Figure 4.* Elastic budget control configurations across computational budgets for Nemotron Nano v3. The $\mathcal{M}_S \to \mathcal{M}_L$ strategy generally offers the best accuracy–latency tradeoffs (23B → 30B), while 30B → 23B preferred at the highest latency regime. 12B → 30B / 30B → 12B provide low-latency options at the smallest budgets (refer to Appendix D for more details).

*Table 2.* Throughput improvements for Nano v3-Elastic models. Throughput is measured with vLLM at bfloat16 on an H100 GPU.

| Elastic Model Variant | ISL/OSL | Max. Batch Size | Speedup |
|---|---|---|---|
| Nano v3 30BA3.6A | 8192/16384 | 36 | 1× |
| Nano v3 23BA2.8A | 8192/16384 | 108 | 1.8× |
| Nano v3 12BA2A | 8192/16384 | 224 | 2.4× |

## 4.5. Cost Efficiency Analysis

**Training efficiency:** A key advantage of Star Elastic is that it eliminates the exploratory training runs required by prior methods such as Minitron (Muralidharan et al., 2024) and Minitron-SSM (Taghibakhshi et al., 2025), which prune and distill multiple candidate architectures to select the best for final knowledge distillation, incurring token costs that scale linearly with the number of models. In contrast, Star Elastic performs end-to-end, router-guided architecture selection in a single elastic training run, simultaneously optimizing all target budgets. Nested training further provides regularization from the full model, reducing the token requirement even for the final distillation stage. Table 3 compares token requirements for deriving

*Table 3.* Token budget comparison across training from scratch, Minitron-SSM, and our proposed approach.

| Method | Explore | Final | Total |
|---|---|---|---|
| Pretraining | 0B | 40T | 40T |
| Minitron-SSM | 480B | 270B | 750B |
| Elastic | 0B | 110B | **110B** |

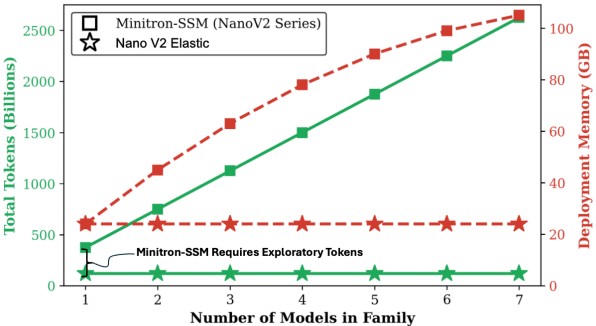

*Figure 5.* Scaling analysis: Star Elastic vs. Minitron-SSM. Star Elastic maintains a constant cost for training tokens and deployment memory as model family size grows, while Minitron-SSM scales linearly[2].

6B and 9B models from a 12B parent (Nano v2). Figure 5 shows Star Elastic achieves constant-cost scaling for both training tokens and deployment memory. Formally, prior methods scale linearly as $\text{Tokens}_{\text{Minitron-SSM}}(n) = n \cdot (\text{Tokens}_{\text{explore}} + \text{Tokens}_{\text{KD}})$, while Star Elastic requires $\text{Tokens}_{\text{Elastic}}(n) = \text{Tokens}_{\text{elastic-KD}} \approx$ constant due to simultaneous multi-budget optimization with shared gradients across nested sub-networks.

**Deployment memory efficiency:** Elastic models with nested weight-sharing offer substantial memory savings: all variants share a single parameter space, with only lightweight routing metadata distinguishing them. Once training is complete, architecture selection can be hard-coded, enabling zero-shot slicing with no runtime overhead. Deploying multiple nested models requires memory equal to the largest model, $\text{Memory}_{\text{Nested}}(n) = \text{Size}(\text{Model}_{\text{max}})$, compared to linear scaling for separate checkpoints, $\text{Memory}_{\text{Separate}}(n) = \sum_{i=1}^{n} \text{Size}(\text{Model}_i)$. For example, elastic Nano v2 (6B, 9B, 12B) fits in 24 GB BF16 versus 54 GB for separate variants, and elastic Nano v3 (12B, 23B, 30B) fits in 60 GB versus 130 GB, highlighting the memory efficiency of Star Elastic's nested models (Table 4, Figure 5).

---

[2]Model family sizes > 3 in Figure 5, as well as Nano v2 6B, Nano v3 23B2.8A, and Nano v3 12B2A in Table 4 are estimated based on scaling laws and have not been trained.

*Table 4.* Deployment memory comparison for Nemotron Nano v2 and V3. Elastic models host multiple budgets in a single nested checkpoint, using less than half the memory of storing separate checkpoints for each variant[2].

| Config | Models | Mem (BF16) |
|---|---|---|
| Elastic (V2) | 6B+9B+12B | **24 GB** |
| NanoV2 | 6B+9B+12B | 54 GB |
| Elastic (V3) | 12B+23B+30B | **60 GB** |
| NanoV3 | 12B+23B+30B | 130 GB |

# 5. Related Work

**Hybrid SSM-Transformer-MoE Models:** Star Elastic focuses on hybrid SSM-Transformer-MoE and SSM-Transformer architectures, both of which have demonstrated strong performance for general tasks and efficient long-context modeling (Gu & Dao, 2023; Dao & Gu, 2024; Lieber et al., 2024; Glorioso et al., 2024; Blakeman et al., 2025a).

**Elastic and Nested Architectures:** MatFormer (Kudugunta et al., 2023) and Flextron (Cai et al., 2024) pioneered nested weight-sharing for Transformers. MatMamba (Shukla et al., 2024) introduces Matryoshka-style sub-block architecture for Mamba layers. FlexGS (Liu et al., 2025) extends this methodology into the computer vision domain. However, to the best of our knowledge, no work supports elastic hybrid Mamba-Attention-MoE architectures, reasoning-focused two-stage training with extended context, or heterogeneous layer-wise architecture selection.

**Reasoning Model Efficiency:** Reasoning-capable LLMs generate extended thought chains for complex problem-solving (Wei et al., 2022; Yao et al., 2023). While prior work focuses on improving reasoning model efficiency via prompting strategies or reinforcement learning (Lightman et al., 2023), we explore a novel axis (model size) to achieve the same objective in this paper.

# 6. Conclusions

We present **Star Elastic**, the first post-training method for producing elastic, reasoning-capable hybrid Mamba–Transformer–MoE and Mamba-Transformer LLMs. Star Elastic efficiently derives 23B and 12B model variants from Nemotron Nano v3 30B, and 9B and 6B variants from Nemotron Nano v2 12B, using only ~150B training tokens, achieving a 360× reduction over training from scratch and a 7× reduction versus sequential compression. Additionally, elastic budget control at inference improves the accuracy–latency trade-off, achieving up to 16% higher accuracy and 1.9× lower latency. All nested models share a constant memory footprint, enabling efficient deployment. Overall, our approach enables state-of-the-art accuracy with adaptive inference and makes training and deploying a family of

models highly cost-effective.

**Limitations and future work:** Our current approach achieves $\sim 3\times$ compression for the smallest model; exploring extreme compression ratios (e.g., $10\times$) for ultra-resource-constrained settings remains an open challenge. Further, task-specific elastic routing (i.e., automatically selecting optimal model configurations based on the input domain, such as code, math, or multilingual tasks) requires further study.

## Impact Statement

This paper presents work whose goal is to advance the field of machine learning. There are many potential societal consequences of our work, none of which we feel must be specifically highlighted here.

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

# A. Router-Selected Architecture Specifications

## A.1. Nemotron Nano v3

The router optimizes for target *active parameter* budgets. Due to the limitation of vLLM (which is used for all benchmark evaluations) in supporting heterogeneous MoE FFN channels, the router is set to select homogeneous configurations across layers.

All variants share 32 attention heads, 64 Mamba heads, 128 MoE experts and the layer pattern. The embedding dimensions and MoE FFN dimensions for each budget are shown in Table 5.

*Table 5.* Nano v3-Elastic architecture variants.

|  | 30B (3.6A) | 23B (2.8A) | 12B (2.0A) |
|---|---|---|---|
| Embedding Dim | 2688 | 2304 | 1920 |
| MoE FFN Dim | 1856 | 1600 | 960 |

**Layer Pattern:** M-E-M-E-M\*-E-M-E-M-E-M\*-E-M-E-M-E-M\*-E-M-E-M-E-M\*-E-M-E-M-E-M\*-E-M-E-M-E-M-E-M-E (M = Mamba, E = MoE, \* = Attention).

## A.2. Nemotron Nano v2

The router optimizes for total parameter budgets and supports heterogeneous FFN dimensions across the 28 dense FFN layers.

All variants share 64 attention heads, 128 Mamba heads and the layer pattern. The embedding dimensions for each budget are shown in Table 6. The 12B variant uses a uniform FFN dimension of 20480 across all layers, while 6B and 9B use heterogeneous configurations detailed below.

*Table 6.* Nano v2-Elastic architecture variants.

|  | 12B | 9B | 6B |
|---|---|---|---|
| Embedding Dim | 5120 | 4608 | 3456 |
| FFN Dim | 20480 | Hetero[†] | Hetero[†] |

[†]Layer-specific dimensions listed below.

**Layer Pattern:** M-M-M-M\*-M-M-M-M\*-M-M-M-M\*-M-M-M-M\*-M-M-M-M\*-M-M-M-M\*-M-M-M-M- (M = Mamba, \* = Attention, - = Dense FFN).

**FFN Dim (6B):** [20480, 9216, 9216, 8192, 9216, 8192, 10240, 9216, 9216, 10240, 9216, 10240, 8192, 10240, 8192, 9216, 9216, 9216, 14336, 8192, 8192, 10240, 7168, 9216, 9216, 8192, 8192, 9216].

**FFN Dim (9B):** [20480, 14336, 14336, 15360, 15360, 14336, 14336, 9216, 13312, 14336, 14336, 15360, 13312, 14336, 14336, 10240, 14336, 8192, 15360, 15360, 14336, 15360, 13312, 9216, 14336, 14336, 8192, 16384].

In both cases, attention and Mamba elastification were omitted in the final models to explore potential cache reuse benefits (see Appendix E).

# B. Nano v2 Benchmark Breakdown

Table 7 shows the breakdown of the benchmark scores for Nano v2 and Nano v2 Elastic (corresponding to the average plot in Figure 3.

# C. Ablation Studies

To validate key design choices, we conduct ablation studies on Nemotron Nano v2. These empirically-derived insights are then applied uniformly to Nano v3, demonstrating that the findings generalize across model families and scales.

## C.1. Effects of Two-Stage Training

Table 8 reveals that Stage 2 extended-context training delivers significant improvements on complex reasoning benchmarks, especially for smaller models. The 6B model gains 19.8% on AIME-2025, while the 12B model gains 4.0%, indicating that smaller models particularly benefit from extended-context adaptation for multi-step reasoning.

## C.2. Impact of Budget Sampling Strategy

Table 9 demonstrates that adjusted sampling ($p(12B) = 0.5, p(9B) = 0.3, p(6B) = 0.2$) substantially improves performance for the full-budget model, particularly on challenging reasoning benchmarks. This confirms that budget-aware curriculum design is essential for balanced multi-target elastic compression.

## C.3. Data Blend Ablation Study

To determine the optimal training data composition for elastic compression, we conduct a comprehensive ablation study across multiple dimensions: data blend ratio (reasoning vs. pretraining data), sequence length, prompt masking strategy, and inclusion of teacher-generated RL samples.

For Nemotron Nano v2, we adopt the data blend ratio established in prior compression studies (Nano, 2025), which identified 70% reasoning (post-training) + 30% pretraining as optimal. We maintain this blend throughout our Nano v2 experiments. For Nemotron Nano v3, no compression study exists in the accompanying report (Blakeman et al., 2025b). We therefore begin with the same 70% reasoning + 30% pretraining blend as our baseline and systematically ablate across additional dimensions: 100% reasoning variants, prompt masking strategies, sequence lengths, augmenting dataset with RL rollouts from the parent model, as detailed below.

All ablations are performed on a 15% embedding-pruned student model derived from Nemotron Nano v3 30B. The

*Table 7.* Detailed Star Elastic results on Nano v2 family. All three variants are obtained from a single 110B-token elastic run. Average accuracy is illustrated in Figure 3. **\*** Indicates teacher model used for distillation.

| Benchmark | NanoV2 Elastic-6B | NanoV2 Elastic-9B | NanoV2 Elastic-12B | NanoV2-9B | NanoV2-12B* | Qwen3-8B |
|---|---|---|---|---|---|---|
| MATH-500 | 96.50 | 97.25 | **97.70** | 97.30 | 97.50 | 96.30 |
| AIME-2024 | 77.64 | 80.26 | **83.44** | 80.89 | 82.90 | 75.83 |
| AIME-2025 | 68.13 | 75.42 | **75.83** | 71.43 | 72.50 | 69.31 |
| GPQA | 53.78 | 62.50 | 63.25 | 63.01 | **65.28** | 59.61 |
| LiveCodeBench v5 | 60.95 | 66.82 | **68.01** | 67.30 | 67.61 | 59.50 |
| MMLU-Pro | 66.65 | 73.45 | 76.20 | 73.61 | **78.47** | 75.50 |

*Table 8.* Two-stage training improvements. Stage 2 (extended-context) provides substantial gains on reasoning benchmarks, particularly AIME-2025, where smaller models benefit significantly (6B: +19.8%, 9B: +9.7%).

| Model (Benchmark) | Stage 1 | Stage 2 | Absolute Gain | Relative Improvement |
|---|---|---|---|---|
| NanoV2 Elastic-6B (MATH-500) | 95.15 | 96.50 | +1.35 | +1.4% |
| NanoV2 Elastic-6B (AIME-2025) | 56.88 | 68.13 | +11.25 | +19.8% |
| NanoV2 Elastic-6B (GPQA) | 49.12 | 53.78 | +4.66 | +9.5% |
| NanoV2 Elastic-9B (MATH-500) | 97.13 | 97.25 | +0.12 | +0.1% |
| NanoV2 Elastic-9B (AIME-2025) | 68.75 | 75.42 | +6.67 | +9.7% |
| NanoV2 Elastic-9B (GPQA) | 59.43 | 62.50 | +3.07 | +5.2% |
| NanoV2 Elastic-12B (MATH-500) | 97.27 | 97.70 | +0.43 | +0.4% |
| NanoV2 Elastic-12B (AIME-2025) | 72.92 | 75.83 | +2.91 | +4.0% |
| NanoV2 Elastic-12B (GPQA) | 62.50 | 63.25 | +0.75 | +1.2% |

same calibration data is used for both importance estimation and knowledge distillation across each configuration.

### C.3.1. ABLATION DIMENSIONS

We systematically vary four key dimensions:

- **Data blend ratio:** 100% reasoning (post-training) vs. 70% reasoning + 30% pretraining.

- **Sequence length:** 8K vs. 32K vs. 49K vs. 256K tokens.

- **Prompt masking:** With prompt masking vs. without prompt masking.

- **RL augmentation:** With and without teacher-generated RL rollouts.

### C.3.2. STAGE 1: INITIAL CONFIGURATION SELECTION (15B TOKENS)

Table 10 presents results for the first stage of ablation, where all configurations are trained for 15B tokens at their respective sequence lengths.

### C.3.3. STAGE 2: CONTEXT EXTENSION STRATEGY (ADDITIONAL 10B TOKENS)

After identifying the optimal Stage 1 configuration (70% reasoning + 30% pretraining, 8K, no prompt mask, no RL rollouts), we investigate context extension strategies by training for an additional 10B tokens (total 25B). We compare

three approaches: (1) continuing at 8K throughout, (2) extending from 8K to 49K in Stage 2, and (3) training at 49K from the start. Results are shown in Table 11.

### C.3.4. KEY FINDINGS

**Stage 1 findings:**

1. **Sequence length:** 8K achieves the best balance between computational efficiency and performance. While longer contexts (32K, 49K) provide modest gains on specific benchmarks (e.g., AIME-2025), they do not consistently improve average performance and incur significantly higher training costs.

2. **Prompt masking:** Removing prompt masking consistently improves performance by 1.5–2.5 percentage points across all benchmarks, suggesting that allowing the model to attend to prompts during training improves reasoning capability.

3. **Blend ratio:** The 70% reasoning + 30% pretraining blend outperforms 100% reasoning by 0.9 percentage points on average. The inclusion of pretraining data provides beneficial diversity and prevents overfitting to post-training distributions.

4. **RL augmentation:** Adding teacher-generated RL rollouts does not provide consistent improvements and occasionally degrades performance on reasoning-heavy benchmarks, potentially due to distribution mismatch.

**Stage 2 findings (context extension):**

*Table 9.* Budget sampling ablation. Adjusted non-uniform sampling substantially improves 12B accuracy on challenging benchmarks (AIME-2025: +3.54%, GPQA: +2.14%) compared to uniform sampling.

| Model | MATH-500 | | AIME-2025 | | GPQA | |
|---|---|---|---|---|---|---|
| | Uniform | Adjusted | Uniform | Adjusted | Uniform | Adjusted |
| NanoV2 Elastic-6B | 96.40 | 96.50 | 67.71 | 68.13 | 55.30 | 53.78 |
| NanoV2 Elastic-9B | 97.40 | 97.25 | 75.00 | 75.42 | 62.75 | 62.50 |
| **NanoV2 Elastic-12B** | 97.33 | 97.70 | 72.29 | **75.83** | 61.11 | **63.25** |

*Table 10.* Stage 1 data blend ablation (15B tokens). The configuration with 70% reasoning + 30% PT has the best overall average. The effect of sequence length is negligible among 8K, 32K, and 49K, while the performance drops significantly at 256K. Data blend abbreviation: PT = Pretraining, SFT = Supervised Finetuning Reasoning, RL = Reinforcement Learning rollouts.

| Configuration (15B token training) | MATH | AIME | GPQA | LiveCode | MMLU | IFB-P | IFB-I | Avg |
|---|---|---|---|---|---|---|---|---|
| *Pruning-only baseline (0B tokens, no training)* | | | | | | | | |
| 1a   100% reasoning, mask, 0B | 89.50 | 40.83 | 40.09 | 38.10 | 59.41 | 48.43 | 53.07 | 52.78 |
| 1b   70% SFT + 30% PT, no mask, 0B | 89.90 | 40.83 | 41.60 | 38.41 | 59.22 | 44.55 | 47.17 | 51.67 |
| 1c   100% reasoning, no mask, 0B | 89.05 | 39.77 | 44.57 | 41.27 | 59.61 | 46.73 | 50.81 | 53.12 |
| *Masking ablation* | | | | | | | | |
| 2a   100% reasoning, mask, 8K | 96.75 | 82.29 | 63.63 | 63.17 | 74.98 | 53.88 | 56.96 | 70.24 |
| **2c   100% reasoning, no mask, 8K** | 97.20 | 85.42 | 68.31 | 66.03 | 77.04 | 55.91 | 57.67 | **72.51** |
| *Blend ratio ablation (70% vs. 100% reasoning)* | | | | | | | | |
| **3a   70% SFT + 30% PT, no mask, 8K** | 97.30 | 85.11 | 69.19 | 65.71 | 76.94 | 59.01 | 60.60 | **73.41** |
| 3b   100% reasoning, no mask, 8K | 97.20 | 85.42 | 68.31 | 66.03 | 77.04 | 55.91 | 57.67 | 72.51 |
| *Extended sequence length (32K, 49K, 256K)* | | | | | | | | |
| 4a   70% SFT + 30% PT, no mask, 32K | 97.20 | 88.13 | 69.44 | 66.35 | 77.55 | 55.98 | 58.45 | 73.30 |
| **4b   70% SFT + 30% PT, no mask, 49K** | 97.50 | 86.46 | 67.99 | 65.07 | 76.90 | 58.84 | 62.03 | **73.54** |
| 4c   70% SFT + 30% PT, no mask, 256K | 96.75 | 84.58 | 66.41 | 61.26 | 76.58 | 56.46 | 59.58 | 71.66 |
| *RL augmentation ablation* | | | | | | | | |
| 5a   30% RL + 49% SFT + 21% PT, 8K | 97.65 | 84.17 | 67.55 | 63.81 | 76.91 | 59.31 | 61.61 | 73.00 |
| 5b   50% RL + 35% SFT + 15% PT, 8K | 97.50 | 85.21 | 69.19 | 66.66 | 76.93 | 57.14 | 59.58 | 73.17 |
| 5c   70% RL + 21% SFT + 9% PT, 8K | 96.95 | 83.75 | 66.47 | 66.35 | 76.72 | 57.48 | 60.77 | 72.64 |

1. **Staged context extension is superior:** The 8K→49K extension strategy (74.40% average) outperforms both 8K-only (72.37%) and 49K-from-start (73.70%) approaches, achieving the best balance between recovery and long-context capability.

2. **Recovery before extension:** Training at 49K from the beginning yields slightly lower performance than staged extension despite using 2B more tokens (27B vs. 25B). This suggests that the compressed model benefits from initial recovery at shorter contexts before adapting to extended sequences.

### C.3.5. FINAL CONFIGURATION

Based on these findings, we adopt the following two-stage training recipe for all elastic compression experiments:

**Stage 1 (short-context):** 70% reasoning + 30% pretraining, 8K sequence length, no prompt masking

**Stage 2 (extended-context):** 70% reasoning + 30% pretraining, 49K sequence length, no prompt masking

This configuration is applied uniformly to both Nemotron Nano v2 and Nano v3 model families, with total training budgets of 110B tokens (Nano v2: 65B + 45B) and 160B tokens (Nano v3: 100B + 60B).

### C.4. Ablation on Depth vs. Width Compression

As elastic compression can target both depth (layer count) and width (hidden dimensions, expert count) of neural networks, we can compared the two approaches in isolation early in the study.

### C.4.1. DEPTH VS. WIDTH COMPRESSION INVESTIGATION

We conducted preliminary ablations on elastic depth compression (15% layer reduction) and width compression (15%

*Table 11.* Stage 2 context extension ablation (25B total tokens). Each configuration starts from the corresponding Stage 1 checkpoint (Table 10). Extending from 8K to 49K (74.40%) outperforms both 8K-only (72.37%) and 49K-from-start (73.70%) strategies. Data blend abbreviation: PT = Pretraining, SFT = Supervised Finetuning Reasoning, RL = Reinforcement Learning.

| Configuration (extra 10B token training) | MATH | AIME | GPQA | LiveCode | MMLU | IFB-P | IFB-I | Avg |
|---|---|---|---|---|---|---|---|---|
| *8K only (no context extension)* | | | | | | | | |
| 3a→   8K continued, same blend | 97.05 | 82.92 | 67.42 | 64.76 | 77.17 | 57.48 | 59.76 | 72.37 |
| *8K → 49K context extension* | | | | | | | | |
| **3a→**   **49K extension, same blend** | 97.40 | 87.08 | 69.94 | 67.94 | 77.49 | 59.25 | 61.67 | **74.40** |
| 3a→   49K extension, 50% RL + 35% SFT + 15% PT | 97.50 | 86.67 | 68.81 | 66.67 | 77.47 | 59.66 | 62.87 | 74.24 |
| 5b→   49K extension, 50% RL + 35% SFT + 15% PT | 97.20 | 88.33 | 69.26 | 66.35 | 77.15 | 57.35 | 61.79 | 73.92 |
| *49K from start (no staged extension)* | | | | | | | | |
| 4b→   49K continued, same blend | 97.65 | 87.50 | 68.75 | 65.71 | 77.63 | 58.02 | 60.65 | 73.70 |
| 4b→   49K continued, 100% RL | 97.20 | 86.04 | 68.18 | 64.36 | 76.96 | 59.93 | 61.91 | 73.51 |
| 4b→   49K continued, 50% RL + 35% SFT + 15% PT | 97.30 | 84.58 | 68.11 | 67.94 | 76.85 | 57.82 | 60.53 | 73.30 |

reduction across experts, FFN hidden dimensions, and embedding dimensions) on Nemotron Nano v3 base model. Both configurations were trained with 25B tokens of knowledge distillation from the full Nano v3 30B base (pretrained only) model as teacher.

Results in Table 12 reveal important tradeoffs between the two approaches. Width-based compression achieves 98.1% of baseline performance (0.768 vs. 0.783 average), maintaining competitive accuracy across nearly all benchmarks. In contrast, depth compression achieves 95.2% of baseline performance (0.745 vs. 0.783 average), with noticeable degradation on HumanEval and MMLU-Pro ( -5.4% each). Given the superior performance of width-based approaches, we prioritize width-based elastic compression for the main results, though depth compression remains viable for extreme latency-constrained scenarios.

### C.4.2. PRACTICAL IMPLICATIONS

For most deployment scenarios prioritizing accuracy, we recommend width-based elastic compression as implemented in Star Elastic. However, for extreme latency-constrained scenarios where inference speed is paramount, and some accuracy loss is acceptable, depth-based compression via layer skipping can be easily activated as an optional mechanism. The Star Elastic method, as described in Section 2 and Appendix F, supports the layer skipping mechanism.

However, depth reduction is omitted in the final models to explore potential cache reuse benefits (see Appendix E).

## D. Per-Benchmark Elastic Budget Control Results

This appendix provides detailed per-benchmark analysis of the elastic budget control results presented in Figure 1 (right). We evaluate hybrid elastic budget control con-

figurations on Nemotron Nano v3 across six benchmarks: AIME-2025, GPQA, MMLU-Pro, LiveCodeBench v5, and IFBench (prompt-strict and instruction-strict modes).

The evaluated configurations are:

- Baseline: 30B model for both thinking and answering
- 23B-30B: 23B for thinking, 30B for answering
- 30B-23B: 30B for thinking, 23B for answering
- 12B-30B: 12B for thinking, 30B for answering
- 30B-12B: 30B for thinking, 12B for answering

### D.1. Overall Performance and Pareto Frontier

Figure 6 demonstrates that hybrid elastic budget control consistently improves the accuracy vs. speed Pareto frontier compared to vanilla Nemotron Nano v3 budget control across all benchmarks. The elastic configurations achieve higher accuracy at equivalent inference times or faster inference at equivalent accuracy, validating the effectiveness of phase-specific model selection.

### D.2. Configuration-Specific Analysis

Figure 7 provides a detailed breakdown of which configurations are optimal at different operating points. The 23B-30B configuration (23B thinking, 30B answering) achieves the best overall balance, particularly excelling with higher accuracy and moderate inference times. For extremely low-latency scenarios where inference time is minimized, the 12B-30B and 30B-12B configurations provide optimal performance, while high latency scenarios prefer the 30B-23B configuration.

### D.3. Budget Control Length Stats

In this section, we analyze the token distribution across three distinct components: prompt, thinking process, and final answer. Table 13 details the average token lengths for

*Table 12.* Depth vs. width compression comparison with 15% parameter reduction and 25B token knowledge distillation. Star Elastic width compression achieves 98.1% of baseline performance (0.768 vs. 0.783 average). Elastic depth compression achieves 95.2% of baseline (0.745 vs. 0.783 average).

| Config | ARC | GSM8K | HumanEval | MATH | MBPP | MMLU-Pro | MMLU | RACE | Wino | Avg |
|--------|-----|-------|-----------|------|------|----------|------|------|------|-----|
| Baseline | 0.918 | 0.915 | 0.703 | 0.775 | 0.724 | 0.613 | 0.782 | 0.884 | 0.762 | 0.783 |
| Depth (15%) | 0.883 | 0.892 | 0.649 | 0.728 | 0.666 | 0.559 | 0.727 | 0.868 | 0.737 | 0.745 |
| Width (15%) | 0.908 | 0.886 | 0.696 | 0.724 | 0.711 | 0.604 | 0.764 | 0.874 | 0.744 | 0.768 |

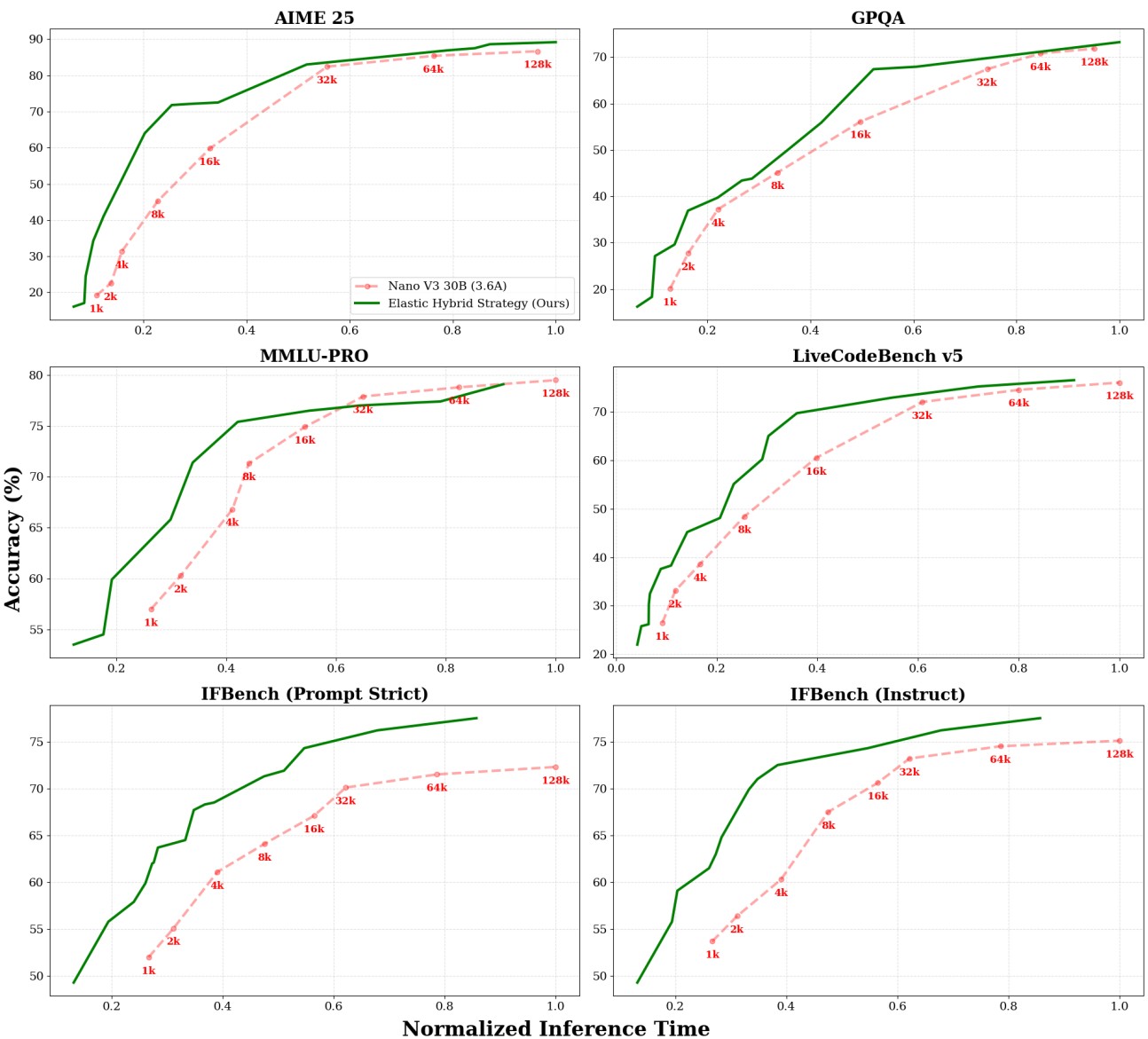

*Figure 6.* Per-benchmark Pareto frontiers comparing hybrid elastic budget control (colored curves) against vanilla Nano v3 budget control (gray). Across all benchmarks, elastic configurations dominate the Pareto frontier, achieving superior accuracy-speed tradeoffs.

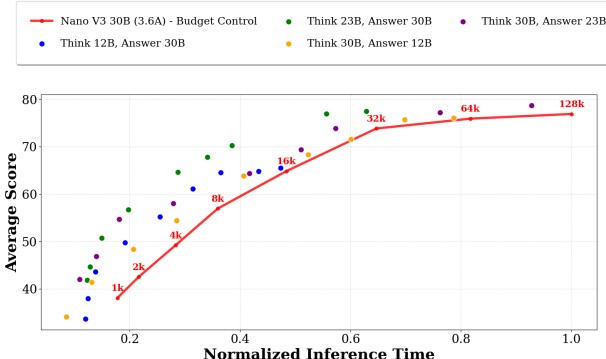

*Figure 7.* Detailed configuration comparison across accuracy-speed space. The 23B-30B configuration dominates at higher scores and moderate inference times, while 12B-30B and 30B-12B are optimal for extremely low-latency scenarios.

each benchmark under a fixed computation budget ($16k$). Note that the average thinking length remains below the budget cap because many samples naturally conclude their generation process before exhausting the full allocation.

*Table 13.* Comparison of budget control statistics. The table shows results for GPQA with a fixed computation budget of $16k$ across different Think–Answer setups.

| Setup | Avg Prompt | Avg Reason | Avg Answer |
|---|---|---|---|
| 23B – 30B | | 11020 | 1066 |
| 12B – 12B | | 12524 | 1291 |
| 12B – 30B | | 12580 | 1270 |
| 23B – 23B | 257 | 10812 | 1062 |
| 30B – 12B | | 10674 | 1106 |
| 30B – 23B | | 10586 | 1033 |
| 30B – 30B | | 10518 | 1035 |

## E. Cache State Reuse

As detailed in Section 3, the nested structure of Star Elastic models preserves the parameters of Mamba and attention layers across different model depths. We hypothesize that this architectural consistency allows for the direct transplantation of KV and SSM state caches from a smaller "thinking" model (e.g., $\mathcal{M}_S$) to a larger "answering" model (e.g., $\mathcal{M}_L$) without the need for recomputation. In this section, we provide empirical evidence validating the structural consistency of these caches and quantify the impact of cache transplantation on generation quality. For all experiments in this section, we use the Nano v2 Elastic family.

To evaluate cache compatibility, we first compare the internal states (specifically, KV caches, Mamba SSM states, and Mamba convolutional states) generated by the nested sub-models against those of the full model, given an identical input sequence. We quantify this alignment by computing

the average cosine similarity across shared layers. As shown in Table 14, we observe high cosine similarity across most state types, particularly for the Mamba SSM states (0.96) and Value caches (0.95), confirming strong structural alignment. While the convolutional states show lower similarity (0.71), their impact is limited due to their relatively small memory footprint (2.2 MB) compared to the dominant SSM states (70 MB). This suggests that the vast majority of the recurrent context is preserved during transplantation.

*Table 14.* Analysis of cache state compatibility between the nested sub-model and the full model given identical input sequences. Memory Footprint indicates the storage requirement for the cache states (BF16 precision). Note the distinct scaling behaviors: Key/Value caches grow linearly (adding 12 KB for every new token), whereas Conv/SSM states have a fixed total size ($O(1)$) that does not increase with sequence length. Cosine similarity is calculated independently for each shared layer and then averaged.

| State Type | Key Cache | Value Cache | Conv State | SSM State |
|---|---|---|---|---|
| Mem. Footprint | 12 KB /tok. | 12 KB /tok. | 2.2 MB | 70 MB |
| Cosine Sim. | 0.89 | 0.95 | 0.71 | 0.96 |

Beyond structural similarity, we evaluate the impact of cache transplantation on downstream task performance using the GSM8k benchmark. Specifically, we use the compressed 6B sub-model to process the prompt and generate the cache states (KV, Conv, and SSM), which are then transplanted to the full 12B model for the answering phase.

Table 15 presents the results. We observe that transplanting the full cache configuration ("All 6B cache") results in a minimal accuracy drop compared to the standard 12B baseline (90.93% vs. 90.18%), a difference likely attributable to generation randomness rather than structural incompatibility. Interestingly, transplanting the convolutional states alone occasionally yields results slightly surpassing the baseline, suggesting that the local features captured by the smaller model are particularly robust. Overall, these results confirm that the transplanted states retain the critical reasoning information required for high-quality generation.

## F. Implementation Details

### F.1. Implementation

The elastic architecture is instantiated through structured masking applied to the hybrid Mamba-Attention-MoE and Mamba-Transformer models. Rather than modifying network topology or creating distinct sub-networks, we apply dimension-specific binary masks that dynamically select active components. This masking-based approach enables efficient training of multiple budgets simultaneously while maintaining architectural transparency and enabling straightforward deployment of any sub-network without architectural recompilation.

*Table 15.* Ablation study of cache transplantation on GSM8k accuracy. The "Baseline" represents standard 12B inference. The "Transplant" rows denote experiments where the specific cache components are generated by the 6B sub-model and transferred to the 12B model for generation. The minimal performance drop in the "All" setting confirms the viability of cache reuse.

| Method | Accuracy |
|---|---|
| 12B Model (Baseline) | 90.93% |
| Transplant 6B KV Cache | 89.26% |
| Transplant 6B Conv State | 91.34% |
| Transplant 6B SSM State | 89.24% |
| Transplant All 6B Cache | 90.18% |

### F.1.1. DYNAMIC MODEL FORMULATION

We present a flexible architecture framework for Star Elastic that enables dynamic adjustment of model dimensions during training through a structured masking approach. Our method builds upon the hybrid Mamba-Attention-MoE and Mamba-Transformer architectures and extends the elastic training paradigm to support comprehensive width and depth flexibility for hybrid architectures.

A dynamic model is obtained by making the stack of layers dynamic and then making each layer type dynamic across different dimensions. If the original LLM is defined as $y = \mathcal{L}_0^N(x)$ where $\mathcal{L}_0^j(x) = \mathcal{L}_0^{j-1}(x) + \mathcal{L}^j(\mathcal{L}_0^{j-1}(x))$, a dynamic layer stack is noted as $\mathcal{D} \circ \mathcal{L}_0^N$ where the operator $\mathcal{D}$ is applied to each layer and makes it dynamic. For example:

$$\mathcal{D} \circ \mathcal{L}^j = (\mathcal{D} \circ \mathcal{L}_j) \cdot \gamma_j \tag{1}$$

where $\gamma_j \in \{0, 1\}$ controls layer retention (depth adaptation) and $\mathcal{D} \circ \mathcal{L}_j$ represents a dynamic Mamba, Attention, FFN, or MoE layer.

The dynamic operator $\mathcal{D}$ applies dimension-specific binary masks $\mathbf{m}$ to the output activations of each layer component, enabling selective feature retention (width adaptation):

$$\mathcal{D}(\mathcal{L}(x)) = \mathcal{L}(x) \odot \mathbf{m} \tag{2}$$

where $\odot$ denotes element-wise multiplication and $\mathbf{m} \in \{0, 1\}^d$ is a binary mask vector that determines which dimensions remain active. Depth adaptation is controlled through the binary coefficient vector $\gamma = [\gamma_0, \gamma_1, \ldots, \gamma_{N-1}]$, while width adaptation is managed through dimension-specific masks applied within each layer type.

### F.1.2. DYNAMIC MAMBA

For Mamba-2 components in the hybrid architecture, we apply group-aware masking following permutation-preserving constraints to maintain structural integrity of state-space computations.

**Dynamic Embedding Mask Operator.** The operator $\mathcal{M}_{\text{emb}}$ applies to any activation or weight matrix with the hidden size $d_e$ as one dimension. For a matrix $\mathbf{W} \in \mathbb{R}^{d_e \times k}$, the masked operation is:

$$\mathcal{M}_{\text{emb}}(\mathbf{W}) = \mathbf{W} \odot (\mathbf{I}_e \otimes \mathbf{1}_k) \tag{3}$$

where $\mathbf{I}_e \in \{0, 1\}^{d_e}$ with $\mathbf{I}_e[0 : i] = 1$ and $\mathbf{I}_e[i + 1 : d_e] = 0$ for some $i \in [0, d_e]$, and $\otimes$ denotes outer product broadcasting across dimension $k$. For matrices $\mathbf{W} \in \mathbb{R}^{k \times d_e}$, the mask broadcasts similarly: $\mathcal{M}_{\text{emb}}(\mathbf{W}) = \mathbf{W} \odot (\mathbf{1}_k \otimes \mathbf{I}_e)$. This operator is applied to layer normalization outputs and all weight matrices interfacing with the embedding dimension.

**Dynamic Mamba Mask Operator.** The operator $\mathcal{M}_{\text{mamba}}$ applies to matrices where dimensions derive from Mamba heads $m_h$ or head channels $m_d$. For a matrix $\mathbf{W} \in \mathbb{R}^{f(m_h, m_d) \times d_e}$ where $f$ represents a dimension function (typically $f(m_h, m_d) = m_h \cdot m_d$), the masked operation is:

$$\mathcal{M}_{\text{mamba}}(\mathbf{W}) = \mathbf{W} \odot (\mathbf{I}_m \otimes \mathbf{1}_{d_e}) \tag{4}$$

where $\mathbf{I}_m \in \{0, 1\}^{f(m_h, m_d)}$ is constructed to satisfy:

$$\mathbf{I}_m[\phi(h, c)] = \begin{cases} 1 & \text{if } h \leq h^* \text{ and } c \leq c^* \\ 0 & \text{otherwise} \end{cases} \tag{5}$$

with $\phi(h, c)$ mapping head $h$ and channel $c$ to flat index, $h^* \in [0, m_h]$ and $c^* \in [0, m_d]$ defining active dimensions. This construction preserves group-aware permutation structure: for heads $h, h' \in \mathcal{G}_g$ belonging to group $g$, $\mathbf{I}_m[\phi(h, \cdot)] = \mathbf{I}_m[\phi(h', \cdot)]$, and maintains head channel consistency: $\mathbf{I}_m[\phi(\cdot, c)]$ is uniform across all heads for each channel $c$.

**Forward Pass.** The dynamic Mamba layer processes input through projection matrices following masked layer normalization. First, we apply the embedding mask to the layer norm output:

$$y_{\text{ln}} = \mathcal{M}_{\text{emb}}(\text{LN}(y)) \tag{6}$$

Then, projections are computed from the masked normalized input:

$$z = \mathbf{W}_z \cdot y_{\text{ln}}, \quad x = \mathbf{W}_x \cdot y_{\text{ln}},$$
$$B = \mathbf{W}_B \cdot y_{\text{ln}}, \quad C = \mathbf{W}_C \cdot y_{\text{ln}}, \quad d_t = \mathbf{W}_{dt} \cdot y_{\text{ln}} \tag{7}$$

where $\mathbf{W}_z, \mathbf{W}_x \in \mathbb{R}^{(m_h \cdot m_d) \times d_e}$, $\mathbf{W}_B, \mathbf{W}_C \in \mathbb{R}^{(g \cdot d_s) \times d_e}$, and $\mathbf{W}_{dt} \in \mathbb{R}^{m_h \times d_e}$. Here, $d_e$ is the embedding dimension, $m_h$ denotes Mamba heads, $m_d$ is the head channel dimension, $g$ represents the number of Mamba groups, and $d_s$ is the SSM state dimension.

We apply the Mamba-specific mask to $z$, $x$, and $d_t$:

$$
\begin{aligned}
z &\leftarrow \mathcal{M}_{\text{mamba}}(z), \\
x &\leftarrow \mathcal{M}_{\text{mamba}}(x), \\
d_t &\leftarrow \mathcal{M}_{\text{mamba}}(d_t)
\end{aligned}
\tag{8}
$$

The intermediate activations $x$, $B$, and $C$ undergo causal convolution:

$$
\hat{x} = \text{conv1d}(x), \quad \hat{B} = \text{conv1d}(B), \quad \hat{C} = \text{conv1d}(C)
\tag{9}
$$

where the conv1d operation on $\hat{x}$ implicitly respects the Mamba mask structure.

The selective state-space model update computes:

$$
\tilde{y} = \text{SSM}(\hat{x}, \hat{B}, \hat{C}, \mathbf{A}, \mathbf{D}, d_t)
\tag{10}
$$

Followed by gated RMSNorm and output projection:

$$
y_{\text{pre}} = \mathbf{W}_O \cdot \text{RMSNorm}(\tilde{y} \odot \text{silu}(z))
\tag{11}
$$

where $\mathbf{W}_O \in \mathbb{R}^{d_e \times (m_h \cdot m_d)}$.

Finally, both dynamic masks are applied to the layer output:

$$
y_{\text{out}} = \mathcal{M}_{\text{emb}}(\mathcal{M}_{\text{mamba}}(y_{\text{pre}}))
\tag{12}
$$

### F.1.3. DYNAMIC ATTENTION

For multi-head attention layers in the hybrid architecture, we apply head-wise and embedding dimension masking to control capacity.

**Dynamic Attention Head Mask Operator.** The operator $\mathcal{M}_{\text{attn\_head}}$ applies to matrices where one dimension derives from attention heads $n_h$ or head dimension $d_h$. For a matrix $\mathbf{W} \in \mathbb{R}^{f(n_h, d_h) \times d_e}$ where $f(n_h, d_h) = n_h \times d_h$, the masked operation is:

$$
\mathcal{M}_{\text{attn\_head}}(\mathbf{W}) = \mathbf{W} \odot (\mathbf{I}_a \otimes \mathbf{1}_{d_e})
\tag{13}
$$

where $\mathbf{I}_a \in \{0, 1\}^{n_h \cdot d_h}$ satisfies:

$$
\mathbf{I}_a[\psi(n, d)] = \begin{cases} 1 & \text{if } n \leq n^* \text{ and } d \leq d^* \\ 0 & \text{otherwise} \end{cases}
\tag{14}
$$

with $\psi(n, d)$ mapping head $n$ and head dimension $d$ to flat index, $n^* \in [0, n_h]$ and $d^* \in [0, d_h]$ defining active dimensions.

**Forward Pass.** The dynamic attention layer processes input through masked layer normalization:

$$
y_{\text{ln}} = \mathcal{M}_{\text{emb}}(\text{LN}(y))
\tag{15}
$$

Projections for query, key, and value are computed as:

$$
\begin{aligned}
\mathbf{Q} &= \mathcal{M}_{\text{attn\_head}}(\mathbf{W}_Q) \cdot y_{\text{ln}}, \\
\mathbf{K} &= \mathcal{M}_{\text{attn\_head}}(\mathbf{W}_K) \cdot y_{\text{ln}}, \\
\mathbf{V} &= \mathcal{M}_{\text{attn\_head}}(\mathbf{W}_V) \cdot y_{\text{ln}}
\end{aligned}
\tag{16}
$$

where $\mathbf{W}_Q, \mathbf{W}_K, \mathbf{W}_V \in \mathbb{R}^{(n_h \cdot d_h) \times d_e}$, with $n_h$ denoting attention heads and $d_h$ the head dimension.

The attention computation follows:

$$
\text{Attn} = \text{softmax}\left(\frac{\mathbf{Q}\mathbf{K}^T}{\sqrt{d_h}}\right)\mathbf{V}
\tag{17}
$$

Followed by output projection:

$$
y_{\text{pre}} = \mathcal{M}_{\text{emb}}(\mathbf{W}_O) \cdot \text{Attn}
\tag{18}
$$

where $\mathbf{W}_O \in \mathbb{R}^{d_e \times (n_h \cdot d_h)}$.

Finally, both dynamic masks are applied to the layer output:

$$
y_{\text{out}} = \mathcal{M}_{\text{emb}}(\mathcal{M}_{\text{attn\_head}}(y_{\text{pre}}))
\tag{19}
$$

### F.1.4. DYNAMIC FFN

This formulation is applicable to both traditional dense FFN and the FFN in each expert in an MoE layer. For these layers, we apply masking to both embedding and intermediate dimensions.

**Dynamic FFN Mask Operator.** The operator $\mathcal{M}_{\text{ffn}}$ applies to matrices where one dimension derives from the FFN intermediate dimension $f_\delta$, where $\delta$ is the layer number corresponding to the current FFN, $1 \leq \delta \leq N_F$. For a matrix $\mathbf{W} \in \mathbb{R}^{f_\delta \times d_e}$, the masked operation is:

$$
\mathcal{M}_{\text{ffn}}(\mathbf{W}) = \mathbf{W} \odot (\mathbf{I}_f \otimes \mathbf{1}_{d_e})
\tag{20}
$$

where $\mathbf{I}_f \in \{0, 1\}^{f_\delta}$ with $\mathbf{I}_f[0:j] = 1$ and $\mathbf{I}_f[j+1:f_\delta] = 0$ for some $j \in [0, f_\delta]$. For matrices $\mathbf{W} \in \mathbb{R}^{d_e \times f_\delta}$, the mask broadcasts similarly.

**Forward Pass.** The dynamic FFN layer processes input through masked layer normalization:

$$
y_{\text{ln}} = \mathcal{M}_{\text{emb}}(\text{LN}(y))
\tag{21}
$$

The first linear transformation with dynamic masking:

$$
h = \mathcal{M}_{\text{ffn}}(\mathbf{W}_1) \cdot y_{\text{ln}}
\tag{22}
$$

where $\mathbf{W}_1 \in \mathbb{R}^{f_\delta \times d_e}$ and $f_\delta$ is the intermediate dimension.

Followed by activation and second linear transformation:

$$
y_{\text{pre}} = \mathcal{M}_{\text{emb}}(\mathbf{W}_2) \cdot \sigma(h)
\tag{23}
$$

where $\mathbf{W}_2 \in \mathbb{R}^{d_e \times f_\delta}$ and $\sigma(\cdot)$ denotes the activation function.

Finally, both dynamic masks are applied to the layer output:

$$y_{\text{out}} = \mathcal{M}_{\text{emb}}(\mathcal{M}_{\text{ffn}}(y_{\text{pre}})) \qquad (24)$$

The complete FFN layer output is thus $\mathcal{D}(\text{FFN}_\delta(y)) = y_{\text{out}}$.

### F.1.5. DYNAMIC MOE

For MoE layers, elasticity is realized by masking experts at the router level rather than modifying expert FFNs directly.

**Dynamic Expert Mask Operator.**  The router produces logits over all experts via a learned linear projection:

$$\mathbf{\Theta} = \mathbf{W}_{\text{router}} \cdot x \in \mathbb{R}^{B \times E}, \qquad (25)$$

where $B$ is the number of tokens and $\mathbf{W}_{\text{router}} \in \mathbb{R}^{E \times d_e}$ is the router weight matrix. Given a target expert count $e(\ell)$ at layer $\ell$, and a global per-layer importance ranking of experts (from REAP (Lasby et al., 2025)), we construct a binary expert mask $\mathbf{m}_{\text{expert}}$ that retains only the top $e(\ell)$ experts:

$$\mathbf{m}_{\text{expert}}[j] = \begin{cases} 1 & \text{if expert } j \text{ is among the top } e(\ell) \\ 0 & \text{otherwise.} \end{cases} \qquad (26)$$

This mask is applied along the expert dimension of the router logits before the routing operation:

$$\mathbf{\Theta}' = \mathbf{\Theta} + \log \mathbf{m}_{\text{expert}}, \qquad (27)$$

where masked experts receive $-\infty$ logits, preventing them from being selected.

**Forward Pass.**  The dynamic MoE layer first computes router logits and applies the expert mask before routing:

$$\begin{aligned} x_{\text{in}} &= \text{LN}(y), \\ \mathbf{\Theta} &= \mathbf{W}_{\text{router}} \cdot x_{\text{in}}, \\ \mathbf{\Theta}' &= \mathbf{\Theta} + \log \mathbf{m}_{\text{expert}}, \end{aligned} \qquad (28)$$

The masked logits $\mathbf{\Theta}'$ are passed to the router's routing function, which performs TopK selection with load balancing:

$$\mathbf{scores}, \mathbf{routing\_map} = \text{routing}(\mathbf{\Theta}'), \qquad (29)$$

The routing function selects the top-$k$ experts per token from the masked logits, applies load-balancing losses (e.g., auxiliary loss, sequence-level load balancing), enforces capacity constraints, and returns routing weights and a routing map indicating token-to-expert assignments. Since only the top $e(\ell)$ experts have finite logits, at most $e(\ell)$ experts can be selected per token, realizing the budget constraint.

### F.1.6. DEPTH ADAPTATION.

Layer-wise depth adaptation is achieved through selective layer retention controlled by $\boldsymbol{\gamma}$. The set of active layers is:

$$\mathcal{A} = \{j \mid \gamma_j = 1, j \in [0, N-1]\} \qquad (30)$$

where $|\mathcal{A}| = N_{\text{target}}$ specifies the target model depth. Skipped layers are bypassed via residual connections:

$$y_{j+1} = \begin{cases} y_j + \mathcal{D} \circ \mathcal{L}_j(y_j) & \text{if } \gamma_j = 1 \\ y_j & \text{if } \gamma_j = 0 \end{cases} \qquad (31)$$

This maintains signal propagation while reducing computation. For hybrid architectures, selective layer retention enables leveraging the complementary strengths of Mamba and attention components at different model scales.

### F.1.7. MASK GENERATION

**Mask Generation from Router Output.**  The router outputs $\pi^{(\text{axis})}$ are processed through Gumbel-Softmax to produce relaxed discrete selections, where $\text{axis} \in \{d_e, m_h, m_d, n_h, e, f\}$. The selected configuration index is determined by $\hat{a}_{\text{axis}} = \arg\max_i \boldsymbol{P}_i^{(\text{axis})}$, where $\boldsymbol{P}^{(\text{axis})}$ is the Gumbel-Softmax probability distribution. In homogeneous mode, if dimension $\text{axis}$ selects configuration index $\hat{a}_{\text{axis}}$, the corresponding target count is $c_{\hat{a}_{\text{axis}}}$ (e.g., number of active embedding channels, depth, or head counts per layer). The binary mask is then constructed by selecting the top $c_{\hat{a}_{\text{axis}}}$ components according to the importance-based ranking $\sigma^{(\text{axis})}$:

$$\mathbf{I}^{(\text{axis})} = \mathbf{I}[\sigma^{(\text{axis})}(j) \leq c_{\hat{a}_{\text{axis}}}], \quad j = 1, \ldots, \text{size}^{(\text{axis})} \qquad (32)$$

In heterogeneous mode, the router output is reshaped into per-layer selections: $\pi^{(\text{axis})}$ is partitioned into $N_X$ segments of size $|\mathcal{X}|$, where each segment determines the configuration for one layer. Per-layer masks are constructed similarly, allowing each layer to have distinct compression ratios. For depth selection, if the router outputs $L_{\text{target}} \in [1, N]$, the top $L_{\text{target}}$ layers from the importance depth ranking are activated via $\gamma_j = 1$ for the selected layers.

The generated masks are then applied to the dynamic model operators $\mathcal{M}_{\text{emb}}, \mathcal{M}_{\text{mamba}}, \mathcal{M}_{\text{attn\_head}}, \mathcal{M}_{\text{ffn}}$, and depth retention coefficients $\boldsymbol{\gamma}$ as defined in the Dynamic Model Formulation section, enabling the model to dynamically adjust capacity.

**Mask Integration Strategies.**  The Gumbel-Softmax probabilities provide differentiable signals for router optimization. We support two mask integration modes:

*Mode 1: Hard Selection via Argmax Logits.* The discrete selection is obtained by $\hat{i}_{\text{axis}} = \arg\max_i P_i^{(\text{axis})}$, and a

hard mask is applied using the corresponding logit:

$$\mathbf{I}_{\text{train}}^{(\text{axis})} = \pi_{\hat{i}_{\text{axis}}}^{(\text{axis})} \cdot \mathbf{I}_{\hat{i}_{\text{axis}}} \tag{33}$$

This directly applies the mask from the selected configuration, scaled by its logit magnitude to provide task-relevant gradient signals.

*Mode 2: Soft Masking via Probabilistic Combination.* Alternatively, masks from all candidate configurations are combined proportionally to their probabilities:

$$\mathbf{I}_{\text{train}}^{(\text{axis})} = \sum_i P_i^{(\text{axis})} \cdot \mathbf{I}_i \tag{34}$$

During training, this soft mask is applied to the dynamic operators, allowing gradients to flow through all configuration options. At inference time, the discrete mask corresponding to $\hat{i}_{\text{axis}}$ from the argmax mode is used and the logit, $\pi_{\hat{i}_{\text{axis}}}^{(\text{axis})}$ is set to 1. We used soft masking throughout this paper.

### F.2. Elastic Model Deployment

A key advantage of the elastic architecture is the ability to extract multiple model variants from a single trained checkpoint without requiring separate training or fine-tuning. This is achieved through a learned slicing mechanism that leverages the router module trained during the elastic training phase.

After training converges, the router has learned optimal budget-aware decisions for every layer and component (attention heads, Mamba, FFN, embeddings). At deployment time, to extract a model for any target budget $B$ that was seen during training, we invoke the router with the budget specification. The router's learned decisions are used to determine which components should be pruned from the full model. These components are then permanently removed (sliced out) from the checkpoint, effectively extracting a nested sub-network that corresponds to the desired parameter count.

Formally, given a trained full model with parameter set $\Theta_{\text{max}}$ and a target budget $B \in \mathcal{B}$ (where $\mathcal{B}$ is the set of budgets used during training), the router $\mathcal{R}$ produces a pruning specification that identifies the parameters to retain. The sliced model parameters are then:

$$\Theta_B = \{\theta \in \Theta_{\text{max}} : \theta \text{ is retained for budget } B\}$$

This zero-shot slicing operation is computationally negligible and produces an inference-ready model immediately, with no retraining, fine-tuning, or additional distillation required. Crucially, any budget $B \in \mathcal{B}$—whether the largest, smallest, or any intermediate size explored during training—can be deployed directly from the single full-model checkpoint.

The practical benefit is substantial: practitioners need to deploy and maintain only a single full-size model checkpoint, yet at inference time can select any of the trained budget variants on-the-fly without cost. This enables dynamic model selection based on per-request latency or resource constraints. Furthermore, all extracted variants share the same learned representations and architectural decisions, ensuring consistency across the model family and eliminating the need for separate fine-tuning or calibration for each size.

