# OpenReview forum: "Star Elastic: Many-in-One Reasoning LLMs with Efficient Budget Control"
_ICML.cc/2026/Conference — ICML 2026 regular_

### Official Review · Reviewer_VuGz · 2026-03-10

**Soundness:** 2
**Presentation:** 3
**Significance:** 2
**Originality:** 2
**Overall Recommendation:** 2
**Confidence:** 2

**Summary:**

This paper proposes Star, an elastic many-in-one model framework that allows a single model to adapt its capacity at inference time based on compute budget. Instead of maintaining multiple models for different deployment constraints, Star uses an elastic architecture with shared parameters and structured subnetworks so one model can operate at multiple scales. Experiments show it can maintain competitive performance across configurations while reducing the number of models and storage cost.

**Compliance With Llm Reviewing Policy:**

Affirmed.

**Final Justification:**

My core concern remains: elastic subnetworks with routing are established techniques, and scaling them to larger hybrid architectures is engineering, not methodological novelty. The paper oversells its conceptual contribution.

**Key Questions For Authors:**

- What is the core difference between the Star method and existing Slimmable/Once-for-All networks?

- During training, do gradient conflicts arise between subnetworks of different scales?

- Does the elastic structure impact inference latency or system throughput?

**Limitations:**

See the weaknesses.

**Strengths And Weaknesses:**

Pros:

- Running several model sizes for different hardware setups is costly to store and maintain. The paper tries to solve this by letting one model handle multiple compute budgets.

- The model can change its effective size by activating different subnetworks, so it can run under different compute limits without needing separate models.

- Results on several configurations suggest the model keeps reasonable performance compared with individually trained models.

Cons:

- The idea of an elastic many-in-one model overlaps with prior work such as Slimmable Networks, Once-for-All Networks, and dynamic width models. The paper does not clearly explain what is fundamentally new in Star.

- The paper relies mostly on experiments and does not explain why a shared model can perform well at different scales or whether parameter sharing causes conflicts or gradient interference.

- Evaluation is done on a small set of model sizes and tasks, without tests on larger models, more benchmarks, or different hardware settings.

- The paper claims reduced model management cost but does not analyze practical metrics like latency, GPU utilization, or system throughput.

---

> ### Author Rebuttal · Authors · 2026-03-30
>
> We thank the reviewer for the feedback. Several concerns appear to reflect points already covered in the paper which we will clarify in the revision.
>
> ## Cons
>
> ### Con 1
>
> We do **not** claim to be the first elastic network architecture overall; Slimmable Networks and Once-for-All are important predecessors. Our contribution is distinct in five ways: (1) **scope**: prior work focuses on CNNs, whereas Star Elastic targets **hybrid Mamba-Attention-MoE LLMs** at **billion-parameter** scale with five elastic axes: embedding dimension, **Mamba/SSM heads**, **MoE**, FFN width, and layer skipping; (2) **routing**: Star uses a differentiable **Gumbel-Softmax router** trained jointly with the model, rather than fixed multipliers or post-hoc prediction; (3) **reasoning-time elasticity**: Star uses different nested submodels for **thinking** and **answering**, improving the accuracy-latency Pareto frontier by up to **16%** in accuracy and **1.9x** in latency; (4) **training**: our two-stage curriculum distillation, from short context to **49K** tokens, is designed for reasoning models; and (5) **scale**: prior elastic methods were shown on image models with millions of parameters, whereas Star Elastic operates on **billion-parameter reasoning LLMs**.
>
> ### Con 2
>
> There is no gradient interference in our approach. As described in Appendix F, each micro-batch is forwarded through exactly **one** subnetwork selected by the router. Elasticity is implemented with binary masks via PyTorch forward hooks, which zero inactive dimensions in Mamba, Attention, MoE, and FFN before the forward pass. Thus, gradients flow only through the active parameters for that configuration; conflicting subnetworks are not optimized simultaneously within one step.
>
> Shared parameters work across scales for two reasons: **importance-based initialization** places the most critical parameters in the shared prefix, and **curriculum-based distillation** trains progressively from larger to smaller subnetworks so smaller models inherit knowledge from the larger shared model. In the revision, we will add **per-submodel training loss curves** showing stable convergence across budgets.
>
> ### Con 3
>
> We respectfully disagree that the evaluation is limited. The paper already covers **10 benchmarks** across math, science, coding, language understanding, instruction following, and agentic reasoning, for **30B/3.6B**, **23B/2.8B**, and **12B/2.0B** active models. We provide the following additional results.
>
> |Model|AA-LCR|SciCode|
> |---|---:|---:|
> |Nemotron Nano 3 30B/3.6B|34.25|32.25|
> |Elastic 30B/3.6B|35.00|32.03|
> |Elastic 23B/2.8B|22.33|25.67|
> |Elastic 12B/2.0B|12.66|13.53|
>
> More hardware settings: we apply post-training FP8 quantization, unlocking more hardware:
>
> |Model|AA-LCR|SciCode|MATH-500|GPQA|AIME|LCB|MMLU-Pro|
> |---|---:|---:|---:|---:|---:|---:|---:|
> |Elastic 30B/3.6B FP8|35.00|32.03|97.35|73.23|86.66|70.16|78.71|
> |Elastic 23B/2.8B FP8|22.33|25.67|97.45|69.05|82.71|65.40|75.42|
> |Elastic 12B/2.0B FP8|11.33|14.57|95.70|57.57|75.83|55.87|68.59|
>
> FP8 roughly halves VRAM, for 128K sequence length, with **30B (3.6B active) ~ 33.6 GB**, **23B (2.8B active) ~ 25.2 GB**, and **12B (2B active) ~ 13.7 GB**, enabling deployment on lower hardware tiers with minimal degradation.
>
> Larger models: We are also applying Star Elastic to a new Nemotron 3 Super variant **75B/9.2B**, elastifying it to **74B/8.1B** (2-in-1 model) active using embedding-dimension and MoE top-K pruning, and training for 20B tokens on 512 H100 GPUs:
>
> |Model|AA-LCR|GPQA|HLE|LCB|MMLU-Pro|SciCode|
> |---|---:|---:|---:|---:|---:|---:|
> |Super Elastic 75B (9.2B active)|58.38|77.02|16.40|77.8|82.02|41.16|
> |Super Elastic 74B (8.1B active)|54.31|75.51|15.10|75.47|81.36|37.87|
>
> The Super elastic variants will be released in the April-May 2026 timeframe.
>
> ### Con 4
>
> These metrics are already in the paper: **latency** in Figure 1 (right), including switching overhead; **throughput** in Table 2 for each nested submodel; and **memory** usage in Table 4. We will make these analyses more prominent in the revision.
>
> ## Questions
>
> **Q1:** See Concern 1. Our novelty is not elasticity in general, but elasticity for hybrid billion-parameter Mamba-Attention-MoE reasoning LLMs, with reasoning-phase inference control, and a far larger scale than prior work.
>
> **Q2:** See Concern 2. Each micro-batch uses exactly one routed subnetwork, so gradients flow only through active parameters. We will add per-submodel loss curves to further document stable convergence.
>
> **Q3:** See Concern 4 and Table 2.
>
> ### Planned revisions
>
> We will:
> 1. Clarify novelty relative to Slimmable Networks and Once-for-All
> 2. Expand the explanation of parameter sharing and the absence of gradient conflicts
> 3. Add the additional benchmark, FP8, hardware, and large-scale results above
> 4. Make latency, throughput, and memory analysis easier to find
> 5. Clarify serving implications, including switching overhead and future cache reuse

---

> > ### Author Rebuttal · Reviewer_VuGz · 2026-04-04
> >
> > On novelty: listing five "distinctions" from prior work does not change the fact that this is incremental engineering. Scaling to larger models is not a new idea. On gradient conflicts: if each micro-batch trains only one subnetwork, this is essentially sequential distillation with routing overhead. The claimed benefit of "joint optimization" is undermined by the authors' own explanation.

---

> > > ### Author Response · Authors · 2026-04-04
> > >
> > > We thank the reviewer for their response, but feel that some of the concerns persist despite being directly addressed in our rebuttal.
> > >
> > > 1.  Optimization Math: The claim that our method is "sequential distillation" is **technically incorrect**. It is true that each micro-batch is training one subnetwork, but we **accumulate gradients across micro-batches within a single global batch**. The shared weights $\theta$ are updated once per step using the combined gradients of all active subnetworks:
> > >     $\theta_{t+1} = \theta_t - \eta \sum_{i \in \text{Subnets}} \nabla L_i(\theta_{i})$. Because $\theta_{t+1}$ is computed using the collective gradients simultaneously, this is **joint optimization**, not sequential training.
> > > 2.  Significant Gains: We are **the first work to propose elastic hybrid reasoning** (using a smaller model for thinking and a larger one for answering and vice versa), which achieves a 1.9x latency speedup and a 16% accuracy improvement. This novelty alone is a significant Pareto frontier gain that goes beyond "incremental engineering" and "distinctions from prior work".
> > > 3.  Addressed Evidence: We previously provided results for a **75B Super Elastic model**, **FP8 hardware metrics**, and all **throughput/latency analysis**. These directly address your concerns, yet they are labeled as "unresolved." We kindly ask the reviewer to acknowledge this data.

---

### Official Review · Reviewer_Pfp5 · 2026-03-12

**Soundness:** 4
**Presentation:** 3
**Significance:** 4
**Originality:** 4
**Overall Recommendation:** 5
**Confidence:** 3

**Summary:**

This paper introduces Star Elastic, a post-training strategy that extracts multiple nested sub-networks from a single parent LLM in one run. Instead of training separate models or iterating through compression, it relies on a trainable router and curriculum-based distillation. Evaluated on hybrid MoE architectures (Nemotron Nano), the extracted submodels maintain competitive performance while saving significant compute. A major use case presented is dynamic budget control during inference, dynamically routing the "thinking" and "answering" phases to different submodels to improve the latency-accuracy trade-off.

**Compliance With Llm Reviewing Policy:**

Affirmed.

**Key Questions For Authors:**

1. How sensitive are the smaller submodels to the choice of the budget sampling distribution during the post-training phase?

2. What are the actual systems requirements for serving Star Elastic models? Specifically, what is the latency penalty for switching between a "thinking" submodel and an "answering" submodel mid-generation in a real-world serving engine?

3. Does the elastic property hold up well if applied to standard dense transformers, or is it highly dependent on the MoE/SSM architectural axes?

**Limitations:**

The authors mention architectural scope limitations, but they should thoroughly address the hardware and software prerequisites for deploying dynamic per-phase routing. Without discussing serving overhead, the claimed latency gains remain somewhat theoretical.

**Strengths And Weaknesses:**

Soundness

Strengths: The end-to-end trainable router combined with a largest-to-smallest curriculum distillation is a solid, empirically validated approach. The benchmarks (GSM8K, MATH, MBPP) confirm that the nested submodels do not degrade compared to standalone training.

Weaknesses: The actual training overhead of the router itself and the hyperparameter sensitivity of the curriculum schedule are largely glossed over. Furthermore, the evaluation is heavily tied to the Nemotron family; testing on more ubiquitous architectures (like Llama 3 or Qwen) would make the empirical claims much stronger.

Presentation

Strengths: The writing is straightforward, and the core concept of elastic budget control for reasoning phases is well-motivated. The Pareto frontier visualizations are particularly effective at communicating the performance trade-offs.

Weaknesses: The mechanics of how the router balances nested constraints across different architectural axes (e.g., SSM vs. MoE pruning) are mostly buried in the appendices. This is critical information that needs to be unpacked in the main text.

Significance

Strengths: Slashing the training costs for model families while enabling per-phase inference routing addresses a major deployment bottleneck for reasoning LLMs. This provides a strong practical recipe for the community.

Weaknesses: The practical deployment of dynamic routing requires specialized serving infrastructure (e.g., seamless submodel swapping or on-the-fly weight masking). The paper mostly ignores the engineering friction required to realize these gains in standard serving engines like vLLM.

Originality

Strengths: Decoupling the "thinking" and "answering" phases via elastic sub-networks is a smart, pragmatic solution to the rigid compute allocation problem in modern reasoning models.

Weaknesses: Dynamic routing, Once-for-All networks, and knowledge distillation are established concepts in vision and early NLP literature. The contribution here is heavily empirical—engineering these existing concepts to scale to billion-parameter MoEs—rather than introducing a fundamentally new algorithmic paradigm.

---

> ### Author Rebuttal · Authors · 2026-03-30
>
> We appreciate the reviewer's points and address them below.
>
> ### Weakness 1
> The router adds negligible overhead: it uses two lightweight MLP layers per layer/axis and is trained jointly with the model in a single end-to-end run. Its behavior is governed by two hyperparameters, kappa (scaling temperature) and tau (Gumbel-Softmax temperature), which control the exploration/exploitation trade-off. With our default settings, the router converges in ~200 steps, negligible relative to and already included in the total training budget. We will add these convergence details and a hyperparameter sensitivity analysis in the camera-ready appendix.
>
> Sensitivity to the budget sampling distribution is already reported in Appendix C.2 and Table 9. During long-context training, uniform sampling hurts the largest model while slightly helping the smallest on some tasks, e.g., GPQA. An adjusted non-uniform distribution that assigns more probability to larger budgets yields a better overall trade-off and especially improves the largest model on challenging reasoning benchmarks.
>
> We plan to apply Star Elastic to the Qwen 3.5 family of models in our future research.
>
> ### Weakness 2
> The router acts over independent axes—embedding dimension, Mamba heads, MoE, FFN width, and layer skipping—with one output head per axis. For each axis, it predicts a distribution over valid choices via Gumbel-Softmax, enabling differentiable soft masking during training and hard selection at inference. The axes are independently conditioned on budget, and the router learns how to allocate budget across axes to minimize distillation loss. These details are currently in Appendix F; we will move the core cross-axis routing mechanism into the main methodology section.
>
> ### Weakness 3
> We address this directly in Question 2 on serving requirements and the latency cost of switching submodels mid-generation.
>
> ### Weakness 4
> We would like to respectfully note that our contributions are not limited to empirical results. Scaling dynamic routing, distillation, and elastic networks to billion-parameter hybrid reasoning LLMs is itself substantial because the challenges differ materially from prior vision or small-NLP settings. Specifically, we: (1) support heterogeneous components—Mamba/SSM, multi-head attention, MoE routing, and dense FFN—within one elastic framework, which prior work does not address; (2) introduce inference-time elastic budget control for reasoning models, where different nested submodels handle different generation phases (thinking vs. answering), improving the accuracy-latency Pareto frontier by up to 16% higher accuracy and 1.9x lower latency; and (3) design a two-stage curriculum, short-context then long-context up to 49K tokens, to preserve reasoning during elastic post-training.
>
> ### Question 1
> Smaller submodels show a clear trade-off with the budget sampling distribution: sampling them more often improves their accuracy but degrades the largest model. On Nemotron Nano v2 with GPQA (Appendix C.2, Table 9), equal sampling (1:1:1) gives 55.30/62.75/61.11 for the 6B/9B/12B submodels, while adjusted sampling (5:3:2), i.e., p(12B)=0.5, p(9B)=0.3, p(6B)=0.2, gives 53.78/62.50/63.25. Thus, favoring larger budgets slightly hurts smaller submodels but improves the largest. Because our goal is to avoid degrading the largest model relative to the original Nemotron Nano v2, we use the adjusted 5:3:2 scheme. We will clarify this in the revision and add accuracy-vs.-training-token plots during 49K-context training for both uniform and adjusted sampling.
>
> ### Question 2
> We are actively developing vLLM support and plan to open-source it. The latency cost of switching between the "thinking" and "answering" submodels mid-generation is already included in Figure 1 (right). For efficient vLLM inference, we are developing kernels that slice tensors on the fly, so switching can occur during prefill, thinking, or answering without added speed penalty. We will add the current overhead numbers to the revision: normalized total wall-clock time is 1.00 for both 23B/30B and 30B/23B switching, with SSM/KV-cache computation 0.10 and 0.08, respectively. TTFT is used as a conservative upper bound for SSM/KV-cache computation and includes prefill, request queuing, and first-token sampling.
>
> ### Question 3
> Although our experiments focus on hybrid Mamba-Attention-MoE models, prior work (LLaMaFlex, Cai et al., ICLR 2025) shows that elastic training also applies to Llama-like dense transformers at the 8B scale. In addition, Nemotron Nano v2 in our paper does not use MoE; it is a hybrid Mamba-Attention model with dense FFN layers and still exhibits the same elastic behavior, showing that MoE is not required. The Mamba/SSM component is a strict superset of attention-only transformers, since an attention-only model can be viewed as a special case, so we do not see hybrid architectures as a limitation. We also plan to extend our study to dense transformer models such as Qwen 3.

---

### Official Review · Reviewer_nWRn · 2026-03-13

**Soundness:** 3
**Presentation:** 3
**Significance:** 3
**Originality:** 3
**Overall Recommendation:** 5
**Confidence:** 3

**Summary:**

This paper aims to address the inefficiency and rigidity of training and deploying LLM families. As compared to the traditional method of training each model variant independently (which incurs lots of compute and storage), the authors proposes Star Elastic that produces nested sub-models in one post-training run. The learnable router determines sub-model architecture and supports elastic budget allocation. The experiments shows impressive 360x reduction in training tokens as compared to from scratch runs, and maximally compresses models by 3x. Overall, the proposed method is efficient, scalable and flexible for practical adoptions.

**Compliance With Llm Reviewing Policy:**

Affirmed.

**Key Questions For Authors:**

1. Does the router generalize well, or does it require retraining/tuning for each new application?
2. Is the “360× token reduction” fair? What is the exact baseline for “training from scratch”? Do savings include the cost of importance estimation, router training, etc.?

**Limitations:**

yes

**Strengths And Weaknesses:**

Strengths

1. The post-training method for hybrid Mamba-Attention-MoE model is novel, enabling simultaneous extraction of multiple nested models from one run.
2. The learnable router gives real-time allocation of compute resources, leading to substantial improvements in both accuracy and latency.
3. The empirical results are comprehensive, and showed significant reduction in training cost and time. The method is also validated across diverse benchmarks.

Weaknesses

1. The framework introduces additional architectural complexity (importance scoring, router, curriculum sampling). Practical adoption may be hard.
2. The paper provides limited analysis of how elastic models generalize to unseen domains, and what failure modes might arise.

---

> ### Author Rebuttal · Authors · 2026-03-30
>
> We appreciate the points brought up by the reviewer, and we have tried our best to address them below.
>
> ### Weakness 1: Additional architectural complexity (importance scoring, router, curriculum sampling) may hinder practical adoption.
>
> We appreciate this concern. We want to emphasize two points: (1) Our implementation is entirely hook-based—elasticity masking is applied through standard PyTorch forward hooks attached to existing modules (MambaMixer, Attention, MLP/MoE) without modifying the original model architecture. Our approach can thus be adopted as a lightweight wrapper over any compatible model. (2) We are in the process of open-sourcing our full implementation, pending internal review.
>
> ### Weakness 2: Limited analysis of how elastic models generalize to unseen domains and what failure modes might arise.
>
> We agree that this is an important direction. Our current evaluation uses the same benchmark suite as the original Nemotron Nano 3 model and spans diverse domains, including mathematics (MATH-500, AIME-2025), coding (LiveCodeBench, SciCode), science (GPQA), language understanding (MMLU-Pro), instruction following (IFBench), and agentic reasoning (Tau2). Across all these domains, nested submodels maintain competitive performance relative to independently trained baselines, suggesting that the elastic property transfers across domains rather than being domain-specific. However, we acknowledge that extreme distribution shifts, such as entirely new modalities or languages unseen during post-training, remain unexplored. We leave evaluation on new languages and additional benchmarks beyond those used for the original Nemotron Nano 3 model to future research.
>
> ### Question 1: Does the router generalize well, or does it require retraining/tuning for each new application?
>
> Generalization across models and budgets: For each model (e.g. Nemotron Nano 3\) and each target budget (e.g. 12B, 23B, 30B), the router finalizes its architecture selection during the short-context training phase and is controlled via two hyperparameters: kappa (the scaling temperature) and tau (the Gumbel-Softmax temperature). If the target budget was seen during training, the router requires no further tuning—it directly maps the budget to the optimal subnetwork configuration.
>
> Unseen models and/or budgets: The router weights are specific to a given elastic model; model-agnostic routing is an open problem which we plan to explore as future work. For new (unseen) budgets, the current framework would require retraining with the new budget included. However, we are actively exploring interpolation of router outputs between trained budgets, which would enable continuous budget selection without retraining. We will clarify this in the revised paper.
>
> ### Question 2: Is the "360x token reduction" fair? What is the exact baseline? Do savings include importance estimation and router training costs?
>
> The 360x reduction is computed relative to pretraining the full model family from scratch (i.e., independently training each model variant from random initialization). We additionally report a 7x reduction compared to Minitron-SSM, which is an established baseline for pruning-and-distillation approaches. Both the importance estimation step and the router training are included in our post-training budget (they are part of the same training run), so the 360x and 7x figures account for all costs of our method.

---

> > ### Author Rebuttal · Reviewer_nWRn · 2026-04-04
> >
> > My concerns are addressed. I will keep my score.

---

### Decision · Program_Chairs · 2026-04-30

**Decision:**

Accept (regular)

**Comment:**

Star Elastic introduces a post-training method that produces multiple nested submodels from a single parent reasoning LLM in one training run, enabling elastic budget control where different submodels handle different inference phases. Two reviewers gave strong accept-level scores. The authors also provided compelling additional results during rebuttal. Reviewer VuGz recommended reject. The reviewer's post-rebuttal response did not engage substantively with the new evidence provided. I therefore weigh this review less heavily. The paper makes a solid contribution in scaling elastic training to billion-parameter hybrid reasoning LLMs with meaningful practical gains.